# ChIP-Hub provides an integrative platform for exploring plant regulome

Liang-Yu Fu[1,2,4], Tao Zhu [1,4], Xinkai Zhou[1,4], Ranran Yu[1,4], Zhaohui He[1], Peijing Zhang[3], Zhigui Wu[1], Ming Chen [3], Kerstin Kaufmann [2✉] & Dijun Chen [1✉]

Plant genomes encode a complex and evolutionary diverse regulatory grammar that forms the basis for most life on earth. A wealth of regulome and epigenome data have been generated in various plant species, but no common, standardized resource is available so far for biologists. Here, we present ChIP-Hub, an integrative web-based platform in the ENCODE standards that bundles >10,000 publicly available datasets reanalyzed from >40 plant species, allowing visualization and meta-analysis. We manually curate the datasets through assessing ~540 original publications and comprehensively evaluate their data quality. As a proof of concept, we extensively survey the co-association of different regulators and construct a hierarchical regulatory network under a broad developmental context. Furthermore, we show how our annotation allows to investigate the dynamic activity of tissue-specific regulatory elements (promoters and enhancers) and their underlying sequence grammar. Finally, we analyze the function and conservation of tissue-specific promoters, enhancers and chromatin states using comparative genomics approaches. Taken together, the ChIP-Hub platform and the analysis results provide rich resources for deep exploration of plant ENCODE. ChIP-Hub is available at https://biobigdata.nju.edu.cn/ChIPHub/.

[1] State Key Laboratory of Pharmaceutical Biotechnology, School of Life Sciences, Nanjing University, Nanjing 210023, China. [2] Department for Plant Cell and Molecular Biology, Institute for Biology, Humboldt-Universität zu Berlin, 10115 Berlin, Germany. [3] Department of Bioinformatics, College of Life Sciences, Zhejiang University, Hangzhou 310058, China. [4]These authors contributed equally: Liang-Yu Fu, Tao Zhu, Xinkai Zhou, Ranran Yu. ✉email: kerstin.kaufmann@hu-berlin.de; dijunchen@nju.edu.cn

Genome-wide charting of transcription factor (TF) binding and epigenetic status has become widely used to study gene-regulatory programs in animals and plants. Chromatin immunoprecipitation sequencing (ChIP-seq) is a powerful method to capture DNA targets for TFs or histone modifications across the entire nuclear genome of any organism[1–7]. From a technical point view, the success of ChIP-seq experiments largely depends on the development and validation of highly gene-specific antibodies or tagged transgenic lines[8–10]. However, crosslinking-based ChIP techniques inherently suffer from several limitations, including low throughput, poor resolution, sub-optimal signal-to-noise ratio, and a tendency to 'detect' false positives[11,12]. In this regard, several recent techniques, such as ChIP-exo[13] and CUT&RUN[14], are alternatives to the current standard of ChIP-seq to improve the resolution in identifying protein binding locations. The in vitro DAP-seq technique[15–17], which is based on screening of a genomic DNA library with an affinity-purified TF followed by high-throughput sequencing, is fast, inexpensive, and more scalable than ChIP-seq for the generation of genome-wide TF binding-site maps. However, only a subset of the TF binding sites identified by DAP-seq is accessible in vivo, and typically individual TFs are analyzed – while in vivo, TFs may interact in a combinatorial, tissue-specific manner with other TFs thereby altering DNA-binding preferences. Complementary in vivo experimental approaches—for example, FAIRE-seq, DNase-seq and ATAC-seq—can identify binding sites in open chromatin regions for all associated factors simultaneously and can provide additional information about DNA-binding proteins and their regulatory functions[8,18]. Thanks to these rapidly developing techniques, a tremendous amount of data have been generated by several large consortia (such as the ENCODE consortium in human[19] and mouse[20], as well as the modENCODE consortium in fly[21] and nematode[22]) or various smaller projects (such as the fruitENCODE project in flowering plants[23]).

Several databases[15,24–28] were recently established for visualization and efficient deployment of public ChIP-seq data by the research community. However, no comprehensive resource is available for plant research. Another major bottleneck in current plant research is the lack of a standardized routine for evaluation and analysis of ChIP-seq data. Therefore, the comparison of data generated by different laboratories is not straightforward, hampering data integration to generate hypotheses for further investigation.

In this work, we comprehensively collect >10,000 public regulatory genomic datasets from >40 plant species and reanalyze them in a uniform way based on the ENCODE standards. All result data are bundled in an integrative platform named ChIP-Hub for visualization and meta-analysis. We explore the co-association of different developmental regulators and associated hierarchical regulatory networks. We provide an atlas of dynamic promoter and enhancer landscapes across representative plant tissues and predict the sequence grammar underlying the chromatin dynamics of tissue-specific regulatory elements. Finally, we apply comparative genomics approaches to investigate the function and conservation of tissue-specific regulatory elements and chromatin states across different plant species.

## Results

**The ChIP-Hub resource**. ChIP-Hub collects all plant regulome data deposited at the NCBI SRA database. These data were generated by high-throughput sequencing experiments including ChIP-seq, DAP-seq, DNase-seq and ATAC-seq. By the time of finalizing this manuscript (as of July 2021), there are >10,000 individual datasets (whose experiment IDs start with SRX, DRX,

or ERX) available at NCBI SRA in >40 plant species, with a nearly exponential growth in recent years (Fig. 1a, b and Supplementary Fig. 1). Although most datasets were generated in model organisms (such as Arabidopsis, rice and maize), the high-throughput regulome experiments have also been widely used in non-model plant species. We manually curated all the datasets through assessing ~540 original publications and >800 biological projects (Fig. 1c) and categorized them into different experimental groups, including open chromatin (11.5%), TFs and other proteins (27.3%), histone-related (39.9%), and input control experiments (19.4%; Fig. 1d).

We adapted the working standards provided by the ENCODE consortium[10] to set up computational pipelines and to systematically reanalyze all public regulome data in plants (Fig. 1e; see Methods). To make our reanalysis results easily accessible to external users, we have developed an integrative web-based platform (ChIP-Hub) to explore all the reanalyzed data sets. Additional data (e.g., sample metadata, references, TF genes, miRNAs, TF motifs, chromatin states and comparative genomics) were also collected and deposited in the database (Fig. 1e). Therefore, the resources are bundled in a well-accessible application that also allows visualization and meta-analyses (Supplementary Fig. 2). Furthermore, in order to continuously add more source data in the future, we have designed ChIP-Hub to be updated quarterly with semi-automatic pipelines, including systematic metadata curation and automatic data processing.

**Comprehensive evaluation of plant regulome data**. One experiment may consist of multiple replicates of ChIP-seq samples and associated control samples. We therefore obtained >6000 individual experiments (Fig. 2a) with manual curation based on experimental designs in original publications or project descriptions. We assigned each experiment to a specific group based on the investigated regulatory factor. ChIP-Hub covers experiments for nearly all plant TF families and well-investigated histone modifications (Fig. 2b and Supplementary Fig. 3).

We then systematically evaluated the data quality of individual experiments ($n = 6055$). Although 89.2% of the experiments have been published in peer-reviewed journals, nearly 40% of the experiments lack control datasets, and only 37.8% have technical or biological replicates (Fig. 2c). Problems of lack of controls or replicates are more obvious in the earlier studies (Supplementary Fig. 4). Nevertheless, most of the evaluated experiments readily meet a variety of quality specifications based on the ENCODE criteria[10] (Fig. 2d). More than 75% of the investigated datasets show moderate to high values of signal-to-noise ratio based on enrichment scores of FRiP (fraction of reads in peaks), NSC/RSC (normalized/relative strand cross-correlation coefficient)[10] and SPOT (signal portion of tags)[29]. Most of the experiments show good quality in terms of the library complexity, as measured by PBC (PCR bottleneck coefficient) and NRF (nonredundant fraction). Not surprisingly, all these quality metrics are positively correlated with each other (Supplementary Fig. 5). As expected, the enrichment SPOT score of experimental groups is significantly higher than that of input control (Fig. 2e). In summary, the above results indicate overall high quality of individual experiments in the analysis.

We identified a total of 52.3 million high-confidence peaks (with an IDR, Irreproducible Discovery Rate[30], <0.05; see Methods) from experiments for open chromatin, annotated TFs and widely-investigated histone H3 modifications (Supplementary Data 1). As expected, peak summits of TF-bound or open chromatin regions generally locate around the transcription start site (TSS) while genomic locations of histone-modified regions vary among different types of histone modifications (Fig. 2f). For

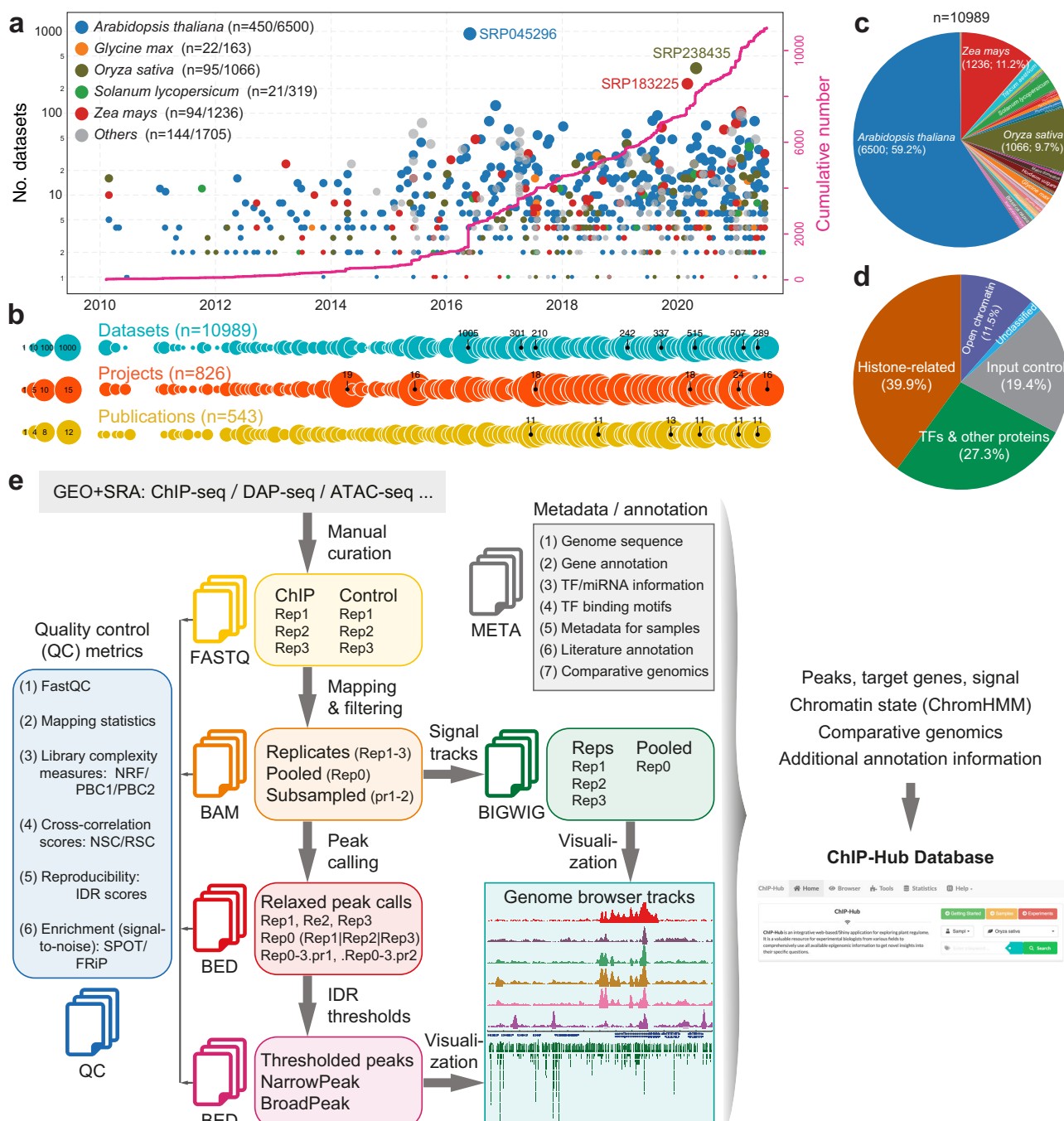

**Fig. 1 The ChIP-Hub platform: data collection and the computational pipeline. a** Explosive generation of regulome and epigenome data in plants. The scatter plot (top) shows the number of datasets over time, as colored by the top representative plant species. Each data point represents one SRA BioProject. The cumulative number is also shown (in pink). **b** Timeline plots showing the overview of the number of datasets, publications and BioProjects over time. **c** Pie chart showing the distribution of datasets by plant species. **d** Pie chart showing the distribution of datasets by sample categories. **e** A standardized, semi-automatic analysis pipeline developed for regulome and epigenome experiments. We adapted the working standards provided by the ENCODE consortium[10] to set up the computational pipeline, including read mapping, peak calling and subsequent statistical treatment of replicates. The resulting data are further integrated by ChromHMM[58] for each plant species. All the metadata as well as analyzed data are bundled in our Shiny application ChIP-Hub for visualization and meta-analysis.

genomes with more than 20 distinct experiments, the number of identified open chromatin regions, TF binding events or histone-modified genomic locations varies from 0.21 million (*Chlamydomonas reinhardtii*; experiments $n = 32$) to 21.4 million (*Arabidopsis thaliana*; $n = 3479$); the fraction of genome associated with TF-bound and histone-modified regions shows an average of 22.0% (Fig. 2g), with comparable proportions found in the mouse

(12.6%) and human (~20%) genomes[19,20]. However, the proportion may be far underestimated for most plant genomes since many regulators have not yet been investigated. Of note, about 3500 individual experiments have been generated in Arabidopsis (Fig. 2b), resulting in annotated genomic regions in terms of chromatin status or TF binding encompassing at least 82.1% of the Arabidopsis genomic sequence in aggregate (Fig. 2g).

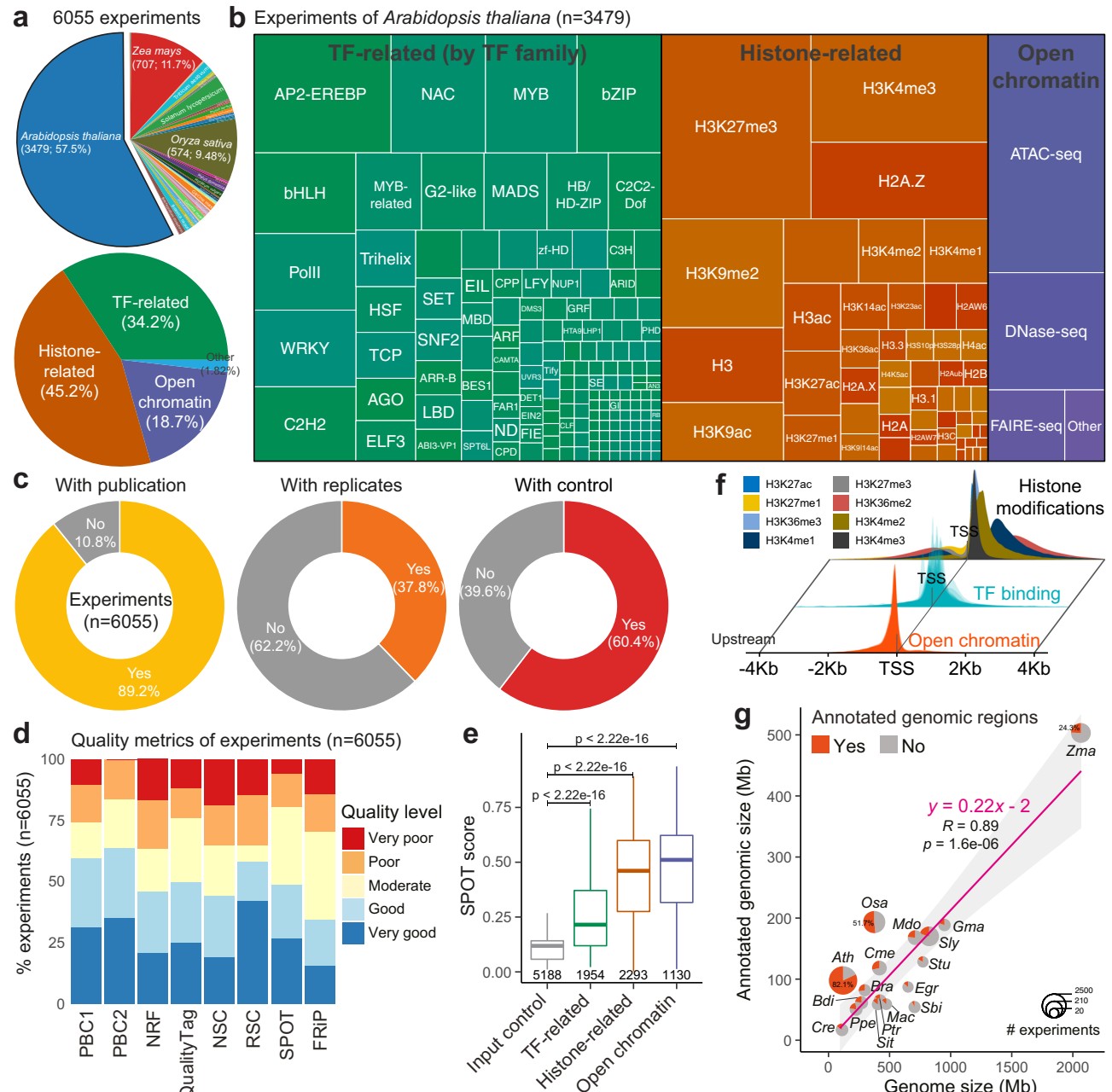

**Fig. 2 Evaluation of plant regulome and epigenome data. a** The annotated experiments by plant species (up) or experimental categories (bottom).
**b** Treemap showing the classification of experiments in *Arabidopsis thaliana* according to transcription factor (TF) families, the types of histone modifications or open chromatin experiments. **c** Donut charts showing different aspects of evaluation of the annotated experiments. **d** The bar chart showing the quality of experiments based on various quality metrics proposed by the ENCODE consortium[10]. **e** Comparison of SPOT scores among different experimental categories. Experiments of input DNA are used for control. The number of datasets in each category is indicated below the boxplot. Statistical significance of difference in terms of the SPOT score between the experiment group and control was calculated by the two-sided Mann–Whitney *U* test. Boxplot shows the median (horizontal line), second to third quartiles (box), and Tukey-style whiskers (beyond the box). **f** Distribution of peak summit around the transcription start site (TSS). **g** Annotated genomic regions versus the genome size. Pie charts show the percentage of genomes annotated by ChIP-seq data. Fitted line and standard errors with 95% confidence intervals are shown. Only genomes with >20 experiments are shown. Full names of genomes can be found in Supplementary Data 11. SPOT: signal portion of tags; FRiP fraction of reads in peaks; NSC normalized strand cross-correlation coefficient. RSC relative Strand cross-correlation coefficient, NRF non-redundant fraction, PBC1/2 PCR bottlenecking coefficients 1/2. Source data are provided as a Source Data file.

Interestingly, 68.8% of Arabidopsis genome is annotated as potential regulatory regions based on 347 ChIP-seq experiments (each has >50 targets) for 157 distinct TFs (Supplementary Fig. 6 and Supplementary Data 2), suggesting pervasive regulatory potential in the compact Arabidopsis genome.

**Extensive TF co-associations and regulatory loops in Arabidopsis.** We investigated TF co-associations and TF-targets gene regulatory networks using TF-related ChIP-seq experiments in Arabidopsis. Integrative analysis of TF-bound genomic regions revealed potential TF co-associations by regulating a similar set of

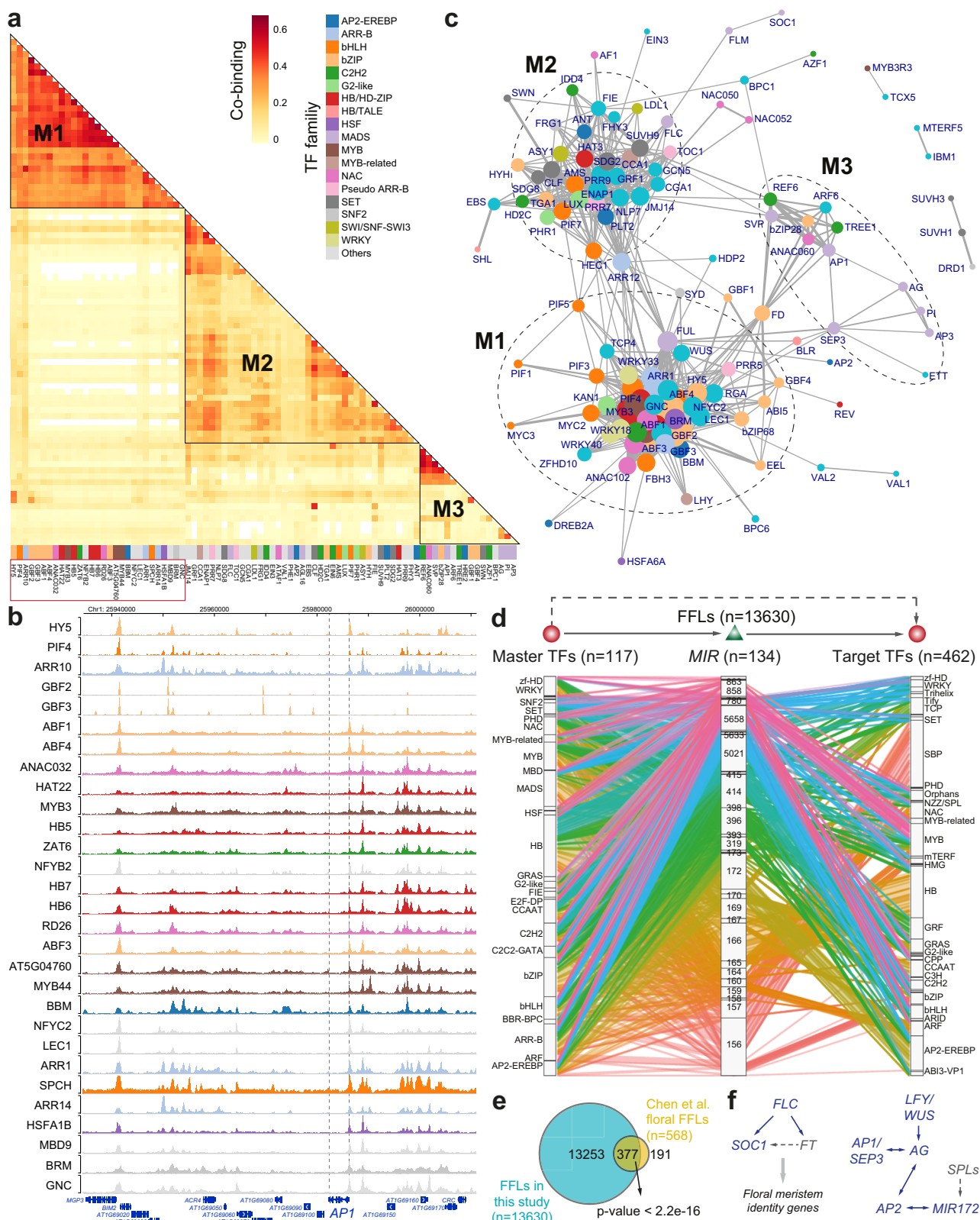

target genes (Fig. 3a and Supplementary Fig. 7), as exemplified around the *APETALA1* (*AP1*) gene locus (Fig. 3b and Supplementary Fig. 8). We organized the pairwise TF co-association into networks with TFs as nodes and their co-binding possibility as edges (Fig. 3c and Supplementary Fig. 9). We observed three dominant co-associated TF modules (M1-M3). M1 consists regulators from TF families of bZIP, bHLH and MYB, while M3

includes MADS TFs response for flower development[31]. Interestingly, M2 contains various regulators for the regulation of histone modifications, including histone acetyltransferases (GENERAL CONTROL NON-REPRESSED PROTEIN5 [GCN5]), deacetylases (HISTONE DEACETYLASE 2 C [HD2C]), methyltransferases (ET DOMAIN GROUP2 [SDG2], FERTILIZATION-INDEPENDENT ENDOSPERM [FIE], SWINGER [SWN] and CURLY LEAF [CLF]), demethylase

**Fig. 3 TF co-associations and hierarchical regulatory networks. a** Co-binding relationships of TFs. Each row or column represent one TF (colored according to its TF family). The significance of co-binding by any two TFs were tested by Jaccard statistics, which measures the ratio of the number of intersecting base pairs occupied by both TFs to the number of base pairs in their union. Three modules (M1-M3) show the highly interplayed regulators. A full co-association heatmap for all investigated TFs ($n = 157$) can be found in Supplementary Fig. 8. **b** Genome browser view of TF binding intensities at the *AP1* locus. Only ChIP-seq experiments for TFs in module M1 are shown. The order of TF ChIP-seq tracks is the same as M1 in **a** (red box). **c** Network showing significant co-associations between TFs. Significant TF co-associations are defined as their co-association scores larger than 0.2, an optimal threshold determined by an elbow statistic (Supplementary Fig. 9a). Three highly interplayed modules in **a** are highlighted. The width of edge represents for the co-association score and the size of node for its degree. **d** Alluvial diagram showing TF-miRNA-TF FFL motifs. Splines were colored based on the family of miRNA genes (*MIR*). The names of TF or miRNA families were labeled. **e** Comparison of FFLs identified in this study and in our previous study[31] based on floral data. The significance of overlap ratio was made by the $\chi^2$ test. **f** Known regulatory loops validated by our predicted FFLs (solid arrows). Regulators without supported ChIP-seq data are colored in grey so that their regulatory interactions are not confirmed (dashed lines). Source data are provided as a Source Data file.

(JUMONJI 14 [JMJ14], RELATIVE OF EARLY FLOWERING 6 [REF6]/JMJ12) and bivalent histone readers (EARLY BOLTING IN SHORT DAY [EBS]) (Fig. 3c). These regulators tightly co-associate with timing regulators such as CIRCADIAN CLOCK ASSOCIATED1 (CCA1)/ LUX ARRHYTHMO (LUX) for the circadian clock and FLOWERING LOCUS C (FLC)/ SHORT VEGETATIVE PHASE (SVP) for the initiation of flowering.

Next, we constructed a hierarchical regulatory network by integrating potential direct TF target genes based on ChIP-seq and predicted miRNA-target interactions, with upstream TFs at the top of the hierarchy, miRNA genes and their target genes at the middle and bottom levels, respectively (Fig. 3d). This "meta-network" includes 117 upstream TFs, 134 miRNA genes, and 462 common target TF genes, and the predicted regulation relationship tends to show a regulator (TF or miRNA) family-specific manner. We identified a comprehensive set of miRNA-mediated feed-forward loops (FFLs; $n = 13,630$) from the meta-network (Fig. 3d and Supplementary Data 3). In particular, the floral FFLs identified in our previous analysis[31] are overrepresented in this extended list ($\chi^2$ test, $p < 2.2e\text{-}16$; Fig. 3e). Furthermore, we validated the confidence of predicted FFLs using known gene interactions from the flowering pathways[32]. For example, regulatory loops involved in the flowering-time regulation[33] and the antagonistic interaction between class A and class C genes in the ABCE model of flower development[34] have been confirmed by our data (Fig. 3f). In sum, the above analysis provides a rich resource to study the biological role of regulatory loops in specific contexts.

**Dynamics of tissue-specific regulatory elements.** More than 1100 open chromatin datasets have been reanalyzed in ChIP-Hub, which offer an opportunity to comprehensively annotate plant regulatory elements such as enhancers. As a proof of concept, we predicted a catalogue of 18,753 promoters and 9976 enhancers in ten representative tissues of Arabidopsis by integrative analysis of 65 open chromatin datasets from nine studies[35–43] (Supplementary Data 4 and 5; see Methods). Clustering analysis based on chromatin accessibility reveals that both promoters and enhancers can distinguish different types of tissues, despite data generated by different studies (Fig. 4a and Supplementary Fig. 10a, b). Supporting this notion, we observed instances of promoters and enhancers that are specifically active in certain types of tissues (Fig. 4b and Supplementary Fig. 11). To compare tissue specificity of promoters and enhancers, we calculated their divergence of chromatin accessibility across tissues based on the Jensen-Shannon diversity (JSD) index. We found that enhancers are generally more tissue-specific than promoters (Fig. 4c). Based on the distribution of JSD score, we defined regulatory elements with JSD > 0.26 as highly specific ones (Fig. 4c). We summarized the number of TF binding sites (including 157 TFs as analyzed above) in both promoters and

enhancers and found that enrichment of TF binding in highly tissue-specific regulatory elements is significantly different between promoters and enhancers (Fig. 4d). However, there is no difference of enrichment of TF binding in low tissue-specific promoters and enhancers.

We further investigated the sequence grammar underlying the chromatin dynamics of tissue-specific regulatory elements. We applied Basset[44] to train a convolutional neural network (CNN) to discriminate one tissue from all other tissues on the basis of the sequence content within accessible sites (Supplementary Fig. 10c, d). The convolutional filters ($n = 600$) in the first CNN layer detect repeatedly occurring local sequence patterns, each comprising a weighted matrix of sequence features *akin to* a TF motif position weight matrix (PWM). The resulting PWMs were matched to known TF motif databases using the Tomtom motif comparison tool[45]. By this we were able to identify known or new motifs represented in tissue-specific promoters and enhancers for each tissue (Fig. 4e and Supplementary Fig. 10e). For example, the classification of root tissues is strongly associated with a filter matching the motif of WOX11, which is critical for initiation of the root primordium during root organogenesis[46,47].

The highly tissue-specific regulatory elements (including 4702 promoters and 4234 enhancers) were grouped into ten different clusters based on their chromatin accessibility (Fig. 5a). We associated potential target genes of these regulatory elements using the 'nearest neighbor' strategy[48], so that one gene may have multiple regulatory elements (Fig. 5b). Note that the nearest neighbor strategy could lead to false target associations due to chromatin loops which can help the formation of interactions between the regulator and its target genes. Nevertheless, the chromatin conformation in Arabidopsis is dominantly represented by kb-sized interactive regions based on Hi-C analyses[49,50], which indicates that enhancers mostly target their neighboring gene(s) in Arabidopsis. Regulatory elements in clusters 2 (C2) and C3 are highly active in flower-related tissues, and their target genes are largely involved in biological processes such as 'flower development' and 'floral organ development' (Supplementary Data 6), including a list of well-known genes controlling floral transition and flower development[32], such as *LEAFY* (*LFY*) *APETALA1* (*AP1*), *FRUITFULL* (*FUL*), *STERILE APETALA* (*SAP*) and *AGAMOUS-LIKE 24* (*AGL24*). Regulatory elements in C4 and C7 are specifically active in root- and leaf-related tissues, with target genes in 'response to biotic stimulus' and 'defense response', respectively. For example, *NAC1* has shown to mediate auxin signaling to promote lateral root development[51], while *YY1* is an important regulator of the ABA response network for plant growth and development[52].

**Evolution of regulatory elements across multiple plant species.** To demonstrate the power of ChIP-Hub for comparative genomic analyses, we mapped the active promoter and enhancer elements

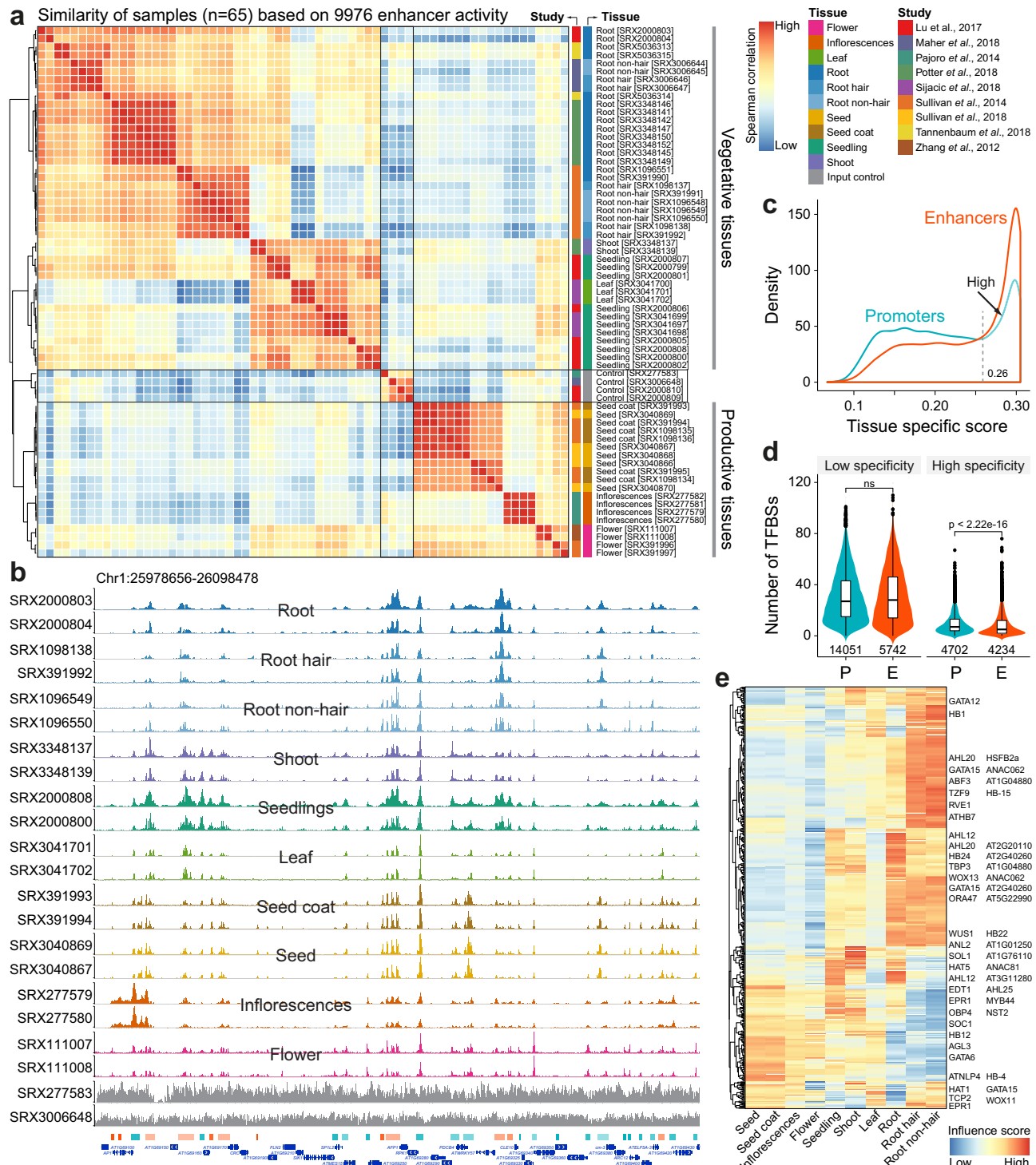

**Fig. 4 Prediction of tissue-specific regulatory elements promoters and enhancers. a** Sample similarity based on enhancer activity. Open chromatin samples (with IDs labeled in square brackets) were collected from nine different studies. The input DNA samples (in grey; $n = 4$) are used for control. Note that samples of productive tissues are well separated from those of vegetative tissues. **b** Genome browser view of selected samples (colored as **a**). Annotated promoters and enhancers are provided at the bottom of tracks. Genome browser view of all samples can be found in Supplementary Fig. 11. **c** Distribution of tissue-specific scores (Jensen-Shannon diversity index) of promoters and enhancers. Highly specific regulatory elements are defined based on a cutoff (0.26) indicated by the dash line. **d** Enrichment of TF binding sites in accessible regions with low or high specificity. P, promoter; E, enhancer; ns, no significance. Statistical significance of difference was calculated by the two-sided Mann–Whitney *U* test. Boxplot showing the median (horizontal line), second to third quartiles (box), and Tukey-style whiskers (beyond the box). **e** Heatmap showing normalized influence of motif-annotated filters on classification of promoters in different tissues. Filters matched to known motifs are labeled. Source data are provided as a Source Data file.

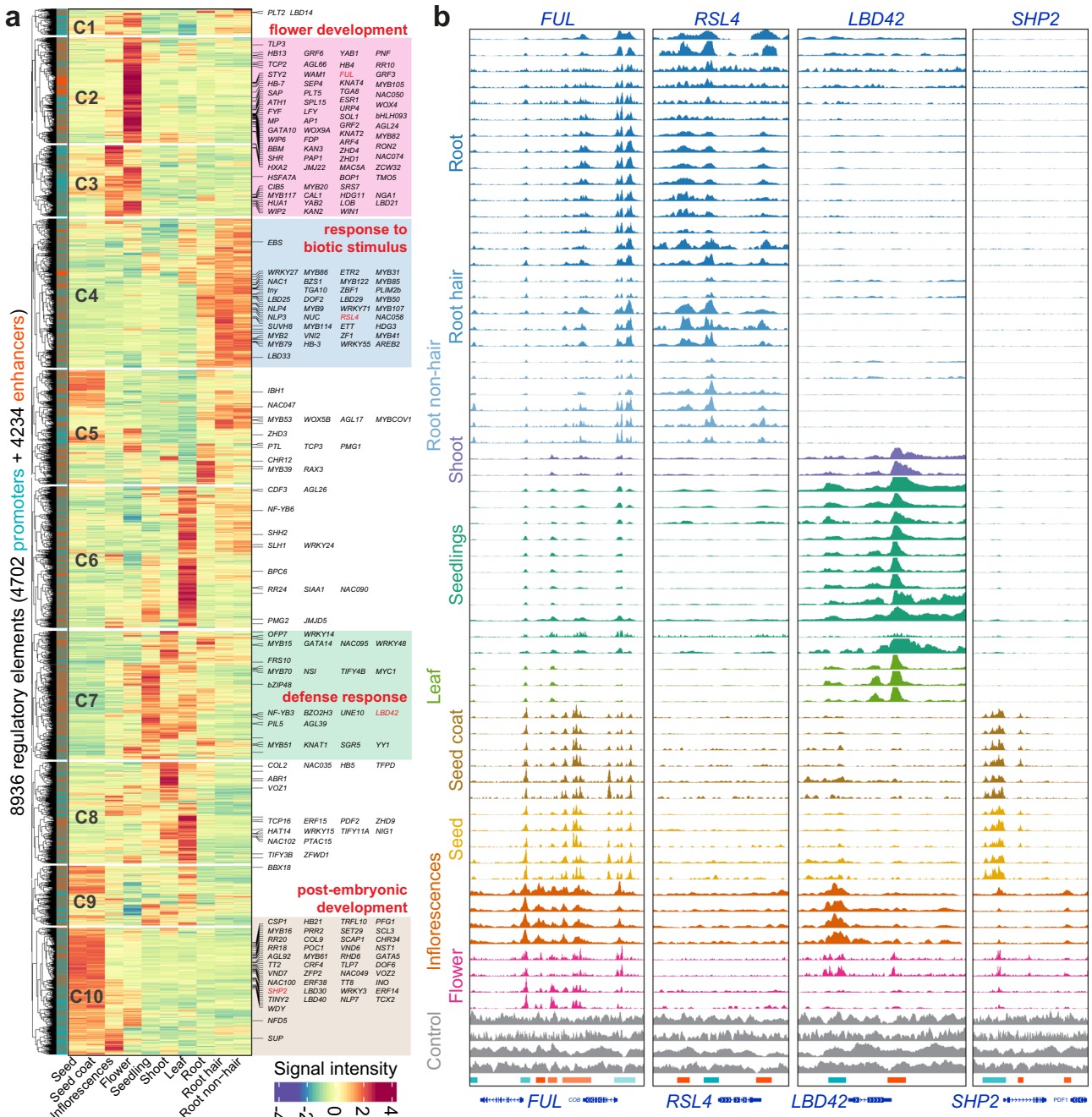

**Fig. 5 Dynamic activity of tissue-specific regulatory elements. a** Heatmap showing the chromatin accessibility of highly specific regulatory elements (including promoters and enhancers). Regulatory elements are grouped into ten clusters (C1–C10; the same number of tissues) based on their activity. TF target genes are labeled on the right. Clusters specific to flower-, root-, leaf- or seed- related tissues are highlighted. Representative of enriched GO terms for the highlighted clusters are indicated. **b** Genome browser views of tissue-specific chromatin accessibility at the four chosen gene loci.

in seedlings across 17 plant species, including five monocots and twelve dicots (Fig. 6a and Supplementary Data 7). A total of 14–28,000 open chromatin regions per species were identified in seedlings using selected ATAC-seq or DNase-seq experiments from ChIP-Hub (Fig. 6b and Supplementary Data 7). Although data used here were taken from different studies, the number of identified peaks was robust to variability in the sample size and the genome size (Supplementary Fig. 12a–c), revealing reliable data quality in our analyses. We predicted promoters and enhancers based on peak data using species-specific criteria since the distribution of distances to the nearest TSS varies largely

among the investigated species (Fig. 6b, c and Supplementary Data 8; see Methods).

We then tracked the evolution of promoters and enhancers across the 17 plant species by pairwise comparison of peaks between species in a reciprocal manner using whole-genome alignments, a similar strategy used for evolutionary analysis of regulatory elements in mammalian species[53]. In brief, conserved promoters or enhancers were defined if the underlying DNA sequence was alignable and the degree of conservation (referred as conservation score hereafter) was considered as the number of species in which the DNA could be aligned (see Methods). We plotted the conserved regulatory elements (including both

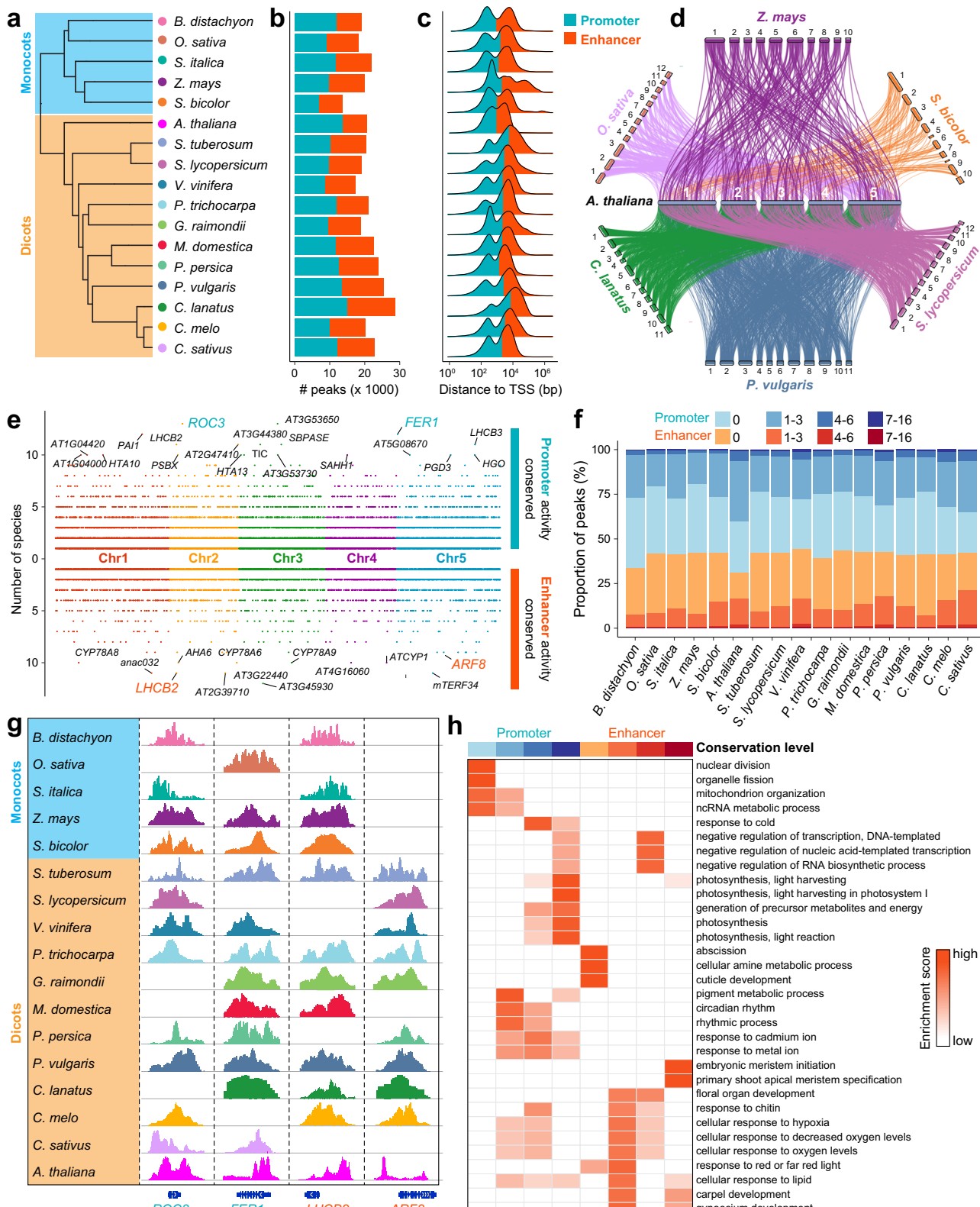

**Fig. 6 Evolutionarily tracking plant promoters and enhancers. a** Phylogenetic tree showing the evolutionary relationships of plant species used in the analysis, including five monocots and twelve dicots. **b** The number of predicted promoters and enhancers in each species. **c** Distance of peak summits to the nearest transcription start site (TSS). **d** Sankey plot showing conserved regulatory elements among seven representative species, using Arabidopsis as a reference. Each line refers active regulatory element (promoter or enhancer) is alignable to the Arabidopsis genome. **e** Dotplots showing the number of species in which the Arabidopsis promoter (above) or enhancer (below) is alignable. Top conserved promoters and enhancers are labeled and four examples are highlighted in (**g**). **f** Barchart summarizing the degree of conservation of promoters and enhancers in each species. **g** Shown are examples of regulatory regions active in different plant species. **h** Enrichment analysis of gene ontology (GO) biological pathways for promoters and enhancers with different degree of conservation. Source data are provided as a Source Data file.

promoters and enhancers) among seven representative species using Arabidopsis as a reference (Fig. 6d). We observed that, as expected, Arabidopsis regulatory elements were generally more conserved within eudicot plant species (*Phaseolus vulgaris*, *Citrullus lanatus* and *Solanum lycopersicum*) than within monocots (*Oryza sativa*, *Zea mays* and *Sorghum bicolor*). We further calculated the conservation score for each promoter and enhancer in Arabidopsis and found that promoters showed a significantly higher degree of conservation than enhancers by ruling out the bias due to sequence alignability between promoters and enhancers (Fig. 6e and Supplementary Fig. 12d–f). This observation suggests that promoter activity is evolutionarily stable while enhancer evolution is relatively more rapid. We performed a similar analysis for promoters and enhancers in other species and found largely consistent results (Fig. 6f).

We highlighted genes associated with highly conserved promoters and enhancers in the Arabidopsis genome (Fig. 6e). As exemplified in Fig. 6g, the genomic regions around the *ROTAMASE CYP 3* (*ROC3*) and *FERRETIN 1* (*FER1*) genes showed conserved promoter activity in most species, and these two genes are both involved in the response to reactive oxygen species[54,55]. The *LIGHT-HARVESTING CHLOROPHYLL B-BINDING 2* (*LHCB2*) gene, which has a conserved function of photosynthesis[56], contained highly conserved enhancers across species. The activities of enhancer regions associated with the *AUXIN RESPONSE FACTOR 8* (*ARF8*) gene, which functions in fruit development in Arabidopsis and tomato[57], appear to be specifically conserved in eudicots. To investigate the functional relevance of promoters and enhancers with different degree of conservation, we performed gene ontology (GO) enrichment analysis for the associated target genes. Interestingly, the most conserved promoters and enhancers were enriched GO terms related to 'meristem development' and 'photosynthesis', while less conserved regulatory elements related to various 'metabolic process' and 'stress response' (Fig. 6h). Overall, the above analyses provide new insights into the plant regulatory genome from an evolutionary aspect.

**Comparison and conservation of tissue-specific chromatin states.** In order to predict the functional relevance of the genomic regions marked by histone modifications, we generated integrated maps of chromatin states in vegetative-, reproductive- or root-related tissues of wide-type plants for genomes with at least five distinct marks (Supplementary Data 9 and Supplementary Fig. 13), using ChromHMM[58] to segment the genome into distinct combinations of histone modification marks (Supplementary Fig. 14). As a proof of concept, a 12-state model was trained in Arabidopsis vegetative-related tissues (Fig. 7a). The resulting "marked" states included six active states, four repressed states and a bivalent state that showed distinct levels of gene expression, chromatin accessibility, TF binding and enrichment for evolutionary conserved noncoding sequences (Fig. 7b–f), accounting for 77.8% of the genome (Fig. 7d) and covering all the major states identified in previous studies[49,59,60]. Particularly, active chromatin states 2 and 3, which are proximal to the TSS, are associated with histone modifications of H3K4me2/H3K4me3 and TF binding for a diverse set of developmental regulators (Fig. 7a, e). State 2 differed from state 3 in that it is enriched with H3K9ac, H3K27ac and H3K36me3 towards TSS-proximal gene body regions. These two states can thus be considered as active promoter states. Interestingly, state 8 is associated with both active mark H3K4me2/H3K4me3 and inactive mark H3K27me3, and enriched with DNA-binding for Polycomb repressive complex 2 (PRC2) and Jumonji proteins, likely being a bivalent or bistable regulatory state[61,62]. This state is highly conserved in

sequences between Arabidopsis and other crucifer species in terms of phastCons score[63] (Fig. 7f). State 9 is a repressed Polycomb state as it is solely associated with H3K27me3 in intergenic regions (Fig. 7a, f). States 10–12 are constitutively enriched with heterochromatin-associated H3K9me2, which is required for the silencing of transposable elements (TEs) and other repetitive DNA[64,65]. The H3K27me3-marked heterochromatin state (state 10) can be facultative as it is enriched with binding for proteins such as nucleosome remodeling complexes and DNA methyltransferases. Overall, our results reveal the previously unappreciated interplay between chromatin state and regulator binding that likely underlies dynamic gene regulation.

The generation of tissue-specific maps of chromatin states (Fig. 7a–f and Supplementary Figs. 15–19) also offers an unprecedented level of comparison of genomic features among different plant species. We thus tracked the evolution of chromatin states in vegetative-related tissues across five plant species (i.e., Arabidopsis, rice, barley, wheat and maize) using Arabidopsis as a reference (see Methods). We observed that most Arabidopsis chromatin states (excepted heterochromatin-related states) were highly conserved in other plant species (Fig. 7g). For example, orthologous sequences were found for 61.1% of Polycomb-repressed regions in at least one of the compared species. Moreover, we found significant epigenomic conservation at orthologous chromatin state-marked regions (Fig. 7h), consistent with results in human[66]. The above findings indicate that plants may share a conserved histone code for gene regulation.

## Discussion

ChIP-seq and complementary assays are powerful methods to measure protein-DNA binding events and chemical modifications of histone proteins at genome-wide level. In recent years, massive research efforts resulted in generation of regulome and epigenome data in various plant species. Re-use and comparison of data from different source studies or among different plant species is not straightforward due to lack of a comprehensive regulome database in the plant field. Given this background, we launched a project in 2015 with an aim of uniform reanalysis and comprehensive evaluation of plant regulome data. Here we provide ChIP-Hub which serves as a comprehensive data portal to explore plant regulomes, especially based on ChIP-seq, DAP-seq and ATAC-seq/DNase-seq experiments. The ChIP-Hub platform is different from several representative plant regulatory element databases, including Plant Cistrome Database[15], ReMap[28], Plant Regulomics[27], PlantRegMap[67], AGRIS[68], JASPAR[69] and CIS-BP[70], in terms of data content, data amount and function specificity. Omics-related datasets collected in ChIP-Hub far exceed data in all those relevant databases. In addition, ChIP-Hub provides predicted TF binding site (TFBS) information using DNA motifs taken from CIS-BP and JASPAR. Most importantly, ChIP-Hub allows comparative regulomic analyses, which provides a unique feature of ChIP-Hub among similar databases.

Although all the evaluated data in ChIP-Hub so far were taken from public databases, unpublished data provided by users can also be analyzed in the same way as published datasets (see online document under the "About" page). In principle, our computational pipeline is easy and ready to adapt to analyze new types of profiling data, such as CUT&RUN experiments for mapping protein-DNA contacts and histone modifications. To this end, a routine to maintain and to update ChIP-Hub in the future has been established. In addition, we are currently improving ChIP-Hub to support the analysis and visualization of plant single cell sequencing data based on ATAC-seq and related techniques.

ChIP-Hub offers a new centralized resource for analysis and comparison of plant regulome and epigenome data. Integrative

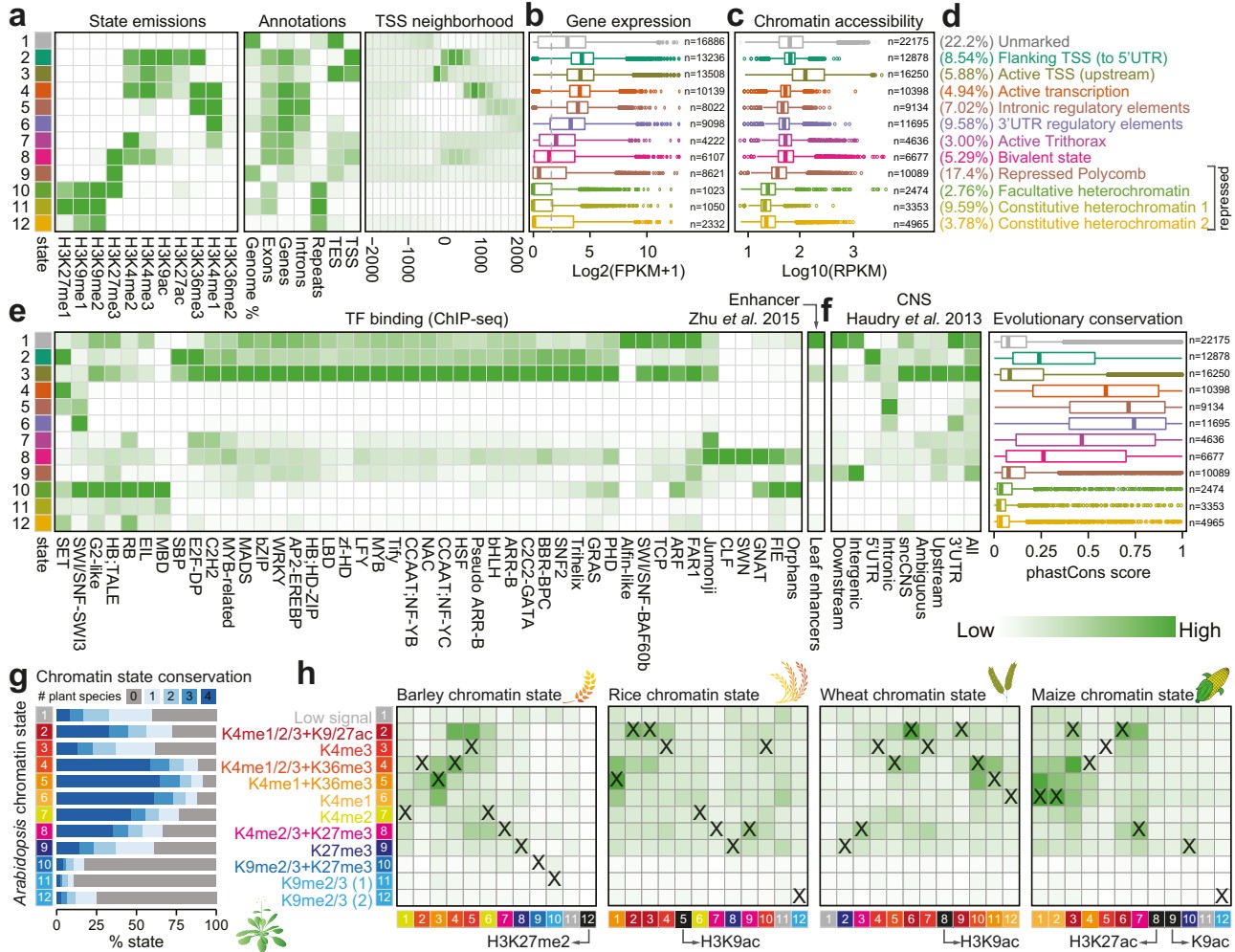

**Fig. 7 Integrative analysis and comparison of chromatin states in plants. a–f** Definition and enrichment for a 12-state ChromHMM[58] model based on eleven histone modification marks in Arabidopsis vegetative-related tissues. Darker green color in the heatmaps indicates a higher probability or enrichment. In the plots, each row corresponds to a different state (in different colors), and each column corresponds to a different mark, a genomic annotation (**a**), gene expression patterns (**b**), chromatin accessibility[42] (**c**), TF binding for a different TF families and leaf enhancers[93] (**e**), or conservation information (**f**). Percentage and description of states summarized based on the overall enrichment of different categories of annotations are shown in **d**. Gene expression data from ref. [93] conserved noncoding sequences (CNSs) and phastCons conservation score (based on nine-way multiple alignment) between Arabidopsis and other crucifers from ref. [63]. Boxplots show the median (horizontal line), second to third quartiles (box), and Tukey-style whiskers (beyond the box). **g, h** Chromatin state conservation between Arabidopsis and other four plant species with annotated states in vegetative (leaves/rosette) tissues. **g** Bar chart showing the percentage of conserved Arabidopsis chromatin states. The number of conserved plants is distinctly colored. Colors for states are explained in **h**. **h** Enrichment of chromatin state conservation between Arabidopsis (row) and other species (column). Pairwise enrichment score was calculated based on Jaccard statistics, which measures the ratio of the number of conserved base pairs to the number of base pairs in union. Darker green in the heatmap indicates a higher enrichment. States with similar compositions of histone modification marks are colored in the same way among different plant species. Matched states between Arabidopsis and other species are labeled as "X". Chromatin states without matched states in Arabidopsis are indicated in black. Unmarked states are colored in grey. Annotation of chromatin states in barley, rice, wheat and maize can be found in Supplementary Figs. 15–19.

analysis of such large-scale datasets using machine-learning-based approaches provides a unprecedented opportunity to extract hidden regulatory genomic patterns and thus to advance our views of a specific biological question under investigation[71,72]. For examples, several integrative studies based on large-scale analysis of TF ChIP-seq data provide a new perspective of gene regulatory networks underlying plant development and evolution[31,73–76]. In our study, we characterized a regulatory landscape across ten different tissues based on open chromatin datasets. We identified an extensive list of tissue-specific regulatory elements (i.e., promoters and enhancers), including those known for tissue-specific gene regulation. We believe that the amounts of data collected in ChIP-Hub will allow

scientists to discover evidences for more specific research points. As an example, we analyzed public DNase-seq data treated by several environmental stresses and identified a set of regulatory elements whose target genes are response to heat shock and dark (Supplementary Fig. 20). The results may provide insights into how plants adapt to changing environments.

As more data are being generated in different plant species, direct comparisons of data among species become possible. As a start point, we tracked the evolution of active functional elements (including promoters and enhancers) across 17 plant species. Consistent with previous findings in mammalian species[53], we observed rapid enhancer and slow promoter evolution in plants (Fig. 6e, f), reflecting a fundamental characteristic of the

regulatory genome. We also have compared the conservation of genomic DNA regions marked by different chromatin states in five plant species with comparable datasets, and found that at least some chromatin states are highly positionally conserved among the investigated species (Fig. 7g, h), suggesting a conserved histone code in plants. In the future, ChIP-Hub would allow to track the evolution of TF binding sites[77–79] and of other active regulatory elements in multiple plant species by comparative genomics.

In summary, we hope that ChIP-Hub will not only allow experimental biologists from various fields to comprehensively use all available regulome and epigenome information to obtain insights into their specific questions, but also allow theoretical biologists to model regulatory relationships under specific conditions and developmental regimes.

## Methods

**Data source, curation and collection**. Metadata of ChIP-seq, DAP-seq, ATAC-seq/DNase-seq samples (equivalent to datasets, accession numbers start with SRX/ERX/DRX) and projects (start with SRP/ERP/DRP) were retrieved from NCBI SRA (https://www.ncbi.nlm.nih.gov/sra), BioSample (https://www.ncbi.nlm.nih.gov/biosample), BioProject (https://www.ncbi.nlm.nih.gov/bioproject) and/or GEO (https://www.ncbi.nlm.nih.gov/geo) databases. ChIP-Hub has a focus on data in "green plants" (i.e., only considering plants in the taxonomy tree with a root ID 33090). Only data generated by Illumina platforms were kept. Firstly, each dataset was associated with publication(s) if available (about 90% of samples can be linked with publications). Then, each dataset was manually curated to determine its investigated factor (i.e., which TF or histone modification mark), its experimental type (whether ChIP or control) and its associated replicates (experiment may have several replicates), based on the metadata and the original publications. Note that it is important to manually check the metadata based on its corresponding publication since some metadata was misannotated in the database. For example, the dataset SRX4063234 in fact contains two different samples, one for ChIP experiment (SRR7142417) and another for control experiment (SRR7142416). In this case, "Run" accessions (start with SRR/ERR/DRR) were instead used as sample accessions (ca. 250 of such cases). For datasets without related publications so far, they were marked as a "unconfirmed" status and would be regularly checked in the future. In general, one experiment may contain replicate samples (i.e., datasets), ChIP sample(s) as well as input control sample(s) and it was designed to investigate regulation of a specific factor (e.g., TF or histone modification) of interest under specific conditions. In the analysis (see the section below), each experiment was processed independently. Furthermore, annotation information for investigated factors was also manually curated. Broadly, factors are grouped into "TFs and other proteins", "histone-related" or "unclassified". For TFs, their gene IDs and family information were also determined if applicable. Finally, a meta file was obtained for each experiment after curation (see Supplementary Data 10 for examples), which is served as an input file for the ChIP-seq computation pipeline (see below).

Raw fastq files for each experiment were downloaded from the European Nucleotide Archive (ENA, https://www.ebi.ac.uk/ena) database. If fastq files were not available at ENA, raw data in the SRA format were downloaded from the SRA database and converted into fastq format using the "fastq-dump" command provided by the SRA Toolkit (version 2.5.1). The "—split-files" option was used for paired-end reads. Fastq files were further checked for completeness before submitted to analysis.

Genome sequences and gene annotations were downloaded from public databases (Supplementary Data 11). Additional annotation data were also included in the ChIP-Hub database in order to better annotate the regulatory factors and their regulatory networks. Annotation for miRNA genes were obtained from miRbase[80] and their genomic locations were updated (by BLAST) based on current reference genomes. TF family information was retrieved from PlantTFDB[81]. TF DNA binding motifs were downloaded from the JASPAR[69], CIS-BP[70] and PlantTFDB[81] databases and were scanned for occurrences in the genome using FIMO[82]. These data were provided as separated data tracks in the genome browser.

**Data processing**. We followed the ChIP-seq data analysis guidelines[10] recommended by the ENCODE project to develop the computational pipeline for various regulome data analyses (Fig. 1e). The analysis pipeline consists of quality control, read mapping, peak calling and assessment of reproducibility among biological replicates and was used to analyze all annotated experiments in a standardized and uniform manner. Specifically, potential adapter sequences were removed from the sequencing reads using the Trim Galore program (version 0.4.1) and the quality of sequencing data was then evaluated by FastQC (http://www.bioinformatics.babraham.ac.uk/projects/fastqc/). Trimmed reads were mapped to the corresponding reference genomes using Bowtie2[83] (version 2.2.6) with parameters "-q—no-unal—threads 8—sensitive". The parameter "-k" was set to 1, 2 and 3 for diploid genomes (e.g., *Oryza sativa*), tetraploid genomes (e.g., *Gossypium*

*barbadense*) and hexaploidy genomes (e.g., *Triticum aestivum*), respectively. Redundant reads and PCR duplicates were removed using Picard tools (v2.60; http://broadinstitute.github.io/picard/) and SAMtools[84] (version 0.1.19).

Peak calling was performed using MACS2[85] (version 2.1.0). Duplicated reads were not considered ("—keep-dup=1") during peak calling in order to achieve a better specificity[86]. The shifting size ("—shift") used in the model was determined by the analysis of cross-correlation scores using the phantompeakqualtools package (https://code.google.com/p/phantompeakqualtools/). The parameter "—call-summits" was used to call narrow peaks. For broad marks of histone modifications (including H3K36me3, H3K20me1, H3K4me1, H3K79me2, H3K79me3, H3K27me3, H3K9me3 and H3K9me1), broad peaks were also called by turning on the "—broad" parameter in MACS2. A relaxed threshold of $p$ value ($p < 1e$-2) was used in order to enable the correct computation of IDR (irreproducible discovery rate) values[10], because IDR requires input peak data across the entire spectrum of high confidence (signal) and low confidence (noise) so that a bivariate model can be fitted to separate signal from noise[30]. Following the recommendations for the analysis of self-consistency and reproducibility between replicates[30], replicate control samples (if available) were combined into one single control in the same experiment. Peak calling was applied to all replicates, pooled data (pooled replicates), pseudo-replicates (half subsample of reads) of each replicate and the pseudo-replicates of pooled sample using the same merged control as input (if applicable). By default, "reproducible" peaks across pseudo-replicates and true replicates with an IDR < 0.05 were recommend for analysis. Besides, peaks with different statistical thresholds are available upon request. For example, "significant" peaks were defined as a fold-change (fold enrichment above background) >2 and a -log10 ($q$ value) >3; while "lenient" peaks as a fold-change >2 and a -log10 ($q$ value) >2. "Relaxed" peaks without additional thresholding were also provided so that any custom threshold can be applied. All peak-based analyses in the pipeline (including peak overlapping, merging and summary) were performed using BEDTools[87] (v2.25.0).

Various metric scores were calculated to assess different aspects of the quality of experiments (https://genome.ucsc.edu/ENCODE/qualityMetrics.html and https://www.encodeproject.org/data-standards/terms/; Fig. 2c, d and Supplementary Fig. 5). For example, library complexity is measured using the non-redundant fraction (NRF) and PCR bottlenecking coefficients 1 and 2 (PBC1 and PBC2). The SPOT (signal portion of tags) score, characterizing the enrichment of signal for each experiment, was calculated by the Hotspot[88] algorithm by subsampling ten million reads. Fraction of reads in peaks (FRiP), another measure of enrichment, is highly correlated with the SPOT score (Supplementary Fig. 5). NSC and RSC (normalized and relative strand cross-correlation coefficient) are related measures of enrichment without dependence on pre-defined peaks, which were calculated by the phantompeakqualtools program[89]. Refer to Supplementary Fig. 21 and Supplementary Data 12 for the definition of metrics categories.

For visualization purpose, wiggle tracks (using pooled data across replicates) were generated by DeepTools[90] with the "bamCoverage" program; different normalization methods (including RPKM [reads per kilobase per million mapped reads], CPM [counts per million mapped reads], BPM [bins per million mapped reads], RPGC [reads per genomic content normalized to 1x sequencing depth] and None) were used to generate different types of signal files. Data signal tracks were visualized in the JBrowse[91] or the WashU Epigenome Browser[92].

**Annotation of promoters and enhancers**. We adopted a similar approach in our previous study[48] to predict Arabidopsis promoters and enhancers using open chromatin data. In Arabidopsis, open chromatin regions whose peak summits are > 1 kb from the nearest TSS were defined as enhancers, and the rest were considered as promoters. Prediction of promoters and enhancers in other plant species was performed in a similar way but with predefined cutoff values (Supplementary Data 7) according to the peak distribution and genome structure of a specific species.

**Assignment of target genes**. Regulatory elements (in layman's terms, called "peaks") were assigned to putative target genes based on the following rules. For a regulatory region overlapping with any gene(s) (protein-coding genes or miRNAs), the overlapping gene(s) were considered as its targets. Otherwise, the regulatory element was assigned to its nearest annotated gene within up to N bp, where N is the median size of intergenic regions (N was set to 3000 if the median size exceeded 3000). The start of genes (i.e., the transcription start site [TSS] of protein-coding genes and the 5' end of miRNA precursors [pre-miRNAs]) was used to calculate the distance. In general, this approach associates a single regulatory element with no more than two genes, with a few exceptions in the case of the regulatory element overlapping multiple genes. This procedure was performed in each species independently.

**Chromatin state analysis**. In order to use the collected histone modification ChIP-seq data from diverse studies for chromatin state analysis and to make the data more comparable among different plant species, only well-characterized H3-related histone modification marks (including H3K9ac, H3K27ac, H3K4me1/2/3, H3K9me1/2/3, H3K27me1/2/3 and H3K36me1/2/3) were considered and only data generated in wild-type plants were used. Furthermore, the datasets were broadly

categorized into vegetative-, reproductive- and root-related samples based on their tissue specificity (Supplementary Data 9). In general, these broadly defined "tissue" types (termed reference tissue types) are more comparative among different plant species and difference in tissue collection by different studies can be eliminated. Although the analysis is cell type agnostic, it is informative even when the relevant cell or tissue type has not been experimentally profiled (this is the most case in plants so far). In addition, we filtered out experiments with less than 1000 called peaks and only considered plant species with at least five distinct types of histone modification marks. In summary, 251 experiments from five plant species were retained for chromatin state analysis (Supplementary Data 9 and Supplementary Fig. 13).

ChromHMM[58] (version 1.19) was applied on the ChIP-seq data of histone modifications in three reference tissue types in five plant species to learn a multivariate HMM model for segmentation of genome in each tissue type. Specifically, the called peaks were first pooled from different ChIP-seq experiments for each type of the histone modifications in each tissue type for each genome separately. Peaks within blacklist regions were excluded from the analysis. The remaining pooled peaks were then processed by the "BinarizeBed" command (with the parameter "-peaks") into binarized data in every 200 bp window over the entire genome. Models were trained independently for each reference tissue type in each genome since the composition of marks varied in different tissue types. We ran the "LearnModel" command with the number of states ranging from 2 states to 15 states and selected an "optimal-state" model based on a rule that the number of states appeared most parsimonious in terms of clearly distinct emission properties and clear interpretability of distinction between states (Supplementary Figs. 15–19). Furthermore, the resulting chromatin states were interpreted based on enrichment analysis of various types of functional annotations, such as gene elements, neighboring gene expression pattern, TF binding, chromatin accessibility and predicted enhancers[48,93]. To this end, the "OverlapEnrichment" and "NeighborhoodEnrichment" commands were used in the analysis. The meaningful mnemonics of states for Arabidopsis vegetative-related tissues was given in Fig. 7d.

**Comparative genomics.** Whole-genome alignments were performed in a way as briefly described below[63]. Firstly, soft masked genomes were aligned to each other using the LastZ alignment algorithm[94]. Collinear alignment blocks separated by gaps of <100 kb were then "chained" according to their locations in both genomes and "netted" to choose the best sub-chain for the reference species[95]. For polyploid plants, each sub-genome was individually analyzed such that each contained non-overlapping chaining. The whole-genome alignments can be visualized together with epigenomic tracks through the integrated Epigenome Browser (see below).

**Cross-species comparisons of regulatory regions.** Pairwise comparisons of regulatory regions (including chromatin states, active promoters and enhancers) were performed by one-to-one mapping annotated regions between species based on the above whole-genome alignments. For regions mapped to multiple orthologous locations in the other genome (i.e., regions split over multiple alignment blocks), only the largest orthologous region in the same alignment block was considered. Marked regions were considered as conserved between species when their orthologous location in the second species overlapped a marked region by a minimum of 50%. Note that the minimum required overlap had little influence on the overall results given that the median value of overlaps is 100% and the mean value is 89.9%.

Conservation scores of regulatory elements were defined as the number of species that a given regulatory region can be alignable to other species and the aligned region showed a peak identified by open chromatin data. To make chromatin state interpretations more comparable across different species (chromatin marks available for state prediction were slightly different among species, see Supplementary Figs. 15–19), the learned chromatin states were re-interpreted (Fig. 7g, h) based on a common set of marks as possible (Supplementary Fig. 22).

**Analysis of gene regulatory networks.** To study gene regulatory networks (GRNs) controlled by TFs with available ChIP-seq data, we focused on a specific network motif, TF-miRNA-TF feed-forward loops (FFLs), which involves targeting of a TF to both miRNAs and miRNA target TFs. Such trifurcate regulatory circuits are of importance for fine tuning of downstream gene expression[96,97]. We highlighted the analysis on Arabidopsis data since a comprehensive list of TFs have been investigated by ChIP-seq experiments in this plant species (Fig. 2b). The methodology, however, can easily be applied to data from any other plants when more and more data are generated. In the miRNA-mediated FFLs, target genes of miRNAs were predicted by the TargetFinder tool[98], with a prediction score cut-off value set to 4. Other relationships (i.e., TF-miRNA and TF-TF) were supported by ChIP-seq data. The final meta-network consisted of regulatory relationships among 117 master TFs, 134 miRNAs and 462 common target TFs (Fig. 3d and Supplementary Data 3), covering nearly two-thirds of the predicted FFLs involved in flower development[31] (Fig. 3e).

**Convolution neural network analysis.** To identify sequence motifs enriched in dynamically accessible regions among different tissues we used Basset[44], a convolutional neural network (CNN) approach. We set the input to the CNN as the 600 bp sequences centered at the summit of the top 2500 highly accessible peaks for each tissue. The output of the classifier is a binary vector of length 10 (i.e., the number of tissue types). We used default Basset values for most parameters, except that we set the number of first layer filters to 600. The CNN contained three convolutional layers, each followed by a rectified linear unit (ReLU) and a max pooling layer, and two fully connected layers. The network architecture is schematically shown in Supplementary Fig. 10c. The predictive performance of the networks was assessed by the *basset_test.lua* script in Basset.

**ChIP-Hub Shiny application.** In order to efficiently use our reanalyzed data by external users, we developed an integrative web-based application (ChIP-Hub) with the Shiny framework (http://shiny.rstudio.com/), which combines the computational power of R with friendly and interactive web interfaces (Supplementary Fig. 2). All the sample metadata, curated metadata and analyzed result data were loaded into a MySQL database, allowing for interactive retrieval through the ChIP-Hub interface. These data were presented in tabular and chart forms in our Shiny web application. Furthermore, the data can be searched by keyword or gene to select datasets of interests. The associated result files, such as wiggle signal files, peak files and additional annotation files, can be loaded into the integrated Epigenome Browser (https://biobigdata.nju.edu.cn/browser/) for visualization.

**Online access and updates.** To make this project easier to maintain for a long life and to update in time, we have developed a semi-automatic computational program (ChIPer) for this purpose. The program regularly (in very month according to our current plan) checks whether any new datasets available in public databases. If so, the new datasets will be sent for curation via email and the curated datasets will be automatically analyzed by the data processing pipeline. New result files will be checked and uploaded to our web server quarterly. Besides, we will include more functionalities in our Shiny application as required.

**Statistical analysis.** If not specified, all statistical analyses and data visualization were done in R (version 3.4.1). R packages such as ggplot2 and plotly were heavily used for graphics. All the sources data for each figure can be found in the Supplementary Information and the newest data can be found in our ChIP-Hub website.

**Reporting summary.** Further information on research design is available in the Nature Research Reporting Summary linked to this article.

## Data availability

ChIP-Hub is available at https://biobigdata.nju.edu.cn/ChIPHub/. Users can view the processed data through the Epigenome Browser https://biobigdata.nju.edu.cn/browser/. All the analysis results (including peak files in the BED format, signal files in the bigwig format, comparative genomics data, predicted promoters/enhancers and gene regulatory networks) can be downloaded through the link https://biobigdata.nju.edu.cn/ChIPHub_download/. Metadata and peak files (in the BED format) for all curated experiments in the current version of ChIP-Hub are also deposited at Zenodo [https://doi.org/10.5281/zenodo.5912234]. Source data are provided with this paper.

## Code availability

The code related to Figures is available at https://biobigdata.nju.edu.cn/ChIPHub_manuscript/.

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

## Acknowledgements

The authors acknowledge the Center for Information Technology and the High Performance Computing Center of Nanjing University and the North-German Supercomputing Alliance (HLRN) for providing high performance computing (HPC) resources that have contributed to the research results reported in this paper. We would like to thank all the data contributors who make this project possible. This work is supported by the National Natural Science Foundation of China (No. 32070656), the Nanjing University Deng Feng Scholars Program and the Priority Academic Program Development (PAPD) of Jiangsu Higher Education Institutions. K.K. wishes to thank the Alexander-von-Humboldt foundation and the Federal Ministry of Education and Research for support. T.Z. appreciate the 2022 Postgraduate Research & Practice Innovation Program of Jiangsu Province (KYCX22_0118). Peijing Zhang and Ming Chen appreciate 2018 Zhejiang University Academic Award for Outstanding Doctoral Candidates, the Fundamental Research Funds for the Central Universities and Collaborative Innovation Center for Modern Crop Production co-sponsored by Province and Ministry. We acknowledge support by the Open Access Publication Fund of Humboldt-Universität zu Berlin.

## Author contributions

D.C. conceived and designed the study. D.C., L.-Y.F., X.Z., R.Y., Z.H., Z.W. and P.Z. annotated the sample metadata. D.C. implemented the computational analysis pipeline and developed the Shiny application with support from X.Z. and M.C. D.C., L.-Y.F. and T.Z. performed the data analyses. D.C. wrote the manuscript with input from K.K. All authors reviewed and approved the submitted version.

## Funding

## Competing interests

The authors declare no competing interests.
