## [Peer Review File · Nature Communications]

ChIP-Hub Provides an Integrative Platform for Exploring Plant RegulomeReviewers' Comments:

Reviewer #1:

Remarks to the Author:

Fu et al. introduced a website ChIP-Hub, used to summarize and display ChIP-seq data in plants. However, many platforms have released similar data sets, providing richer tools for data mining, while the function of ChIP-Hub is rather limited. I did not discover the uniqueness of the platform, nor did it elaborate on specific biological issues. It is just another form of accumulation of existing data and a description of corresponding information. Even if this information is sorted out in great detail, it is of little help for functional research. Nevertheless, I have to admit that the article layout and website design are very good.

Major

1. The only feature of ChIP-Hub is the large amount of data, without in-depth analysis. All data comes from public databases and can be processed within 3-4 weeks without any technical difficulties. Apart from storing public data, I am not sure what the purpose of this platform is.

2. There are several representative plant regulatory element websites, all with specific functions. There are many functional overlaps. Where is the innovation of ChIP-Hub?

1) PlantCistromeDB-Atlases of the Plant Cistrome and Epicistrome:

http://neomorph.salk.edu/dap_web/pages/index.php , which contains TFBS information, and you can conveniently view the combination of different TFs on the genome at the same time on the browser Site.

2) PlantTFDB: <http://planttfdb.gao-lab.org/> , which mainly collects comprehensive information on transcription factors in plants, and provides a convenient entry for researchers to understand the information of transcription factors of interest. It can also predict the transcription factor of a specific gene or protein sequence.

3) Plant regulomics: <http://bioinfo.cemps.ac.cn/plant-regulomics/> , which helps retrieve regulatory elements up and downstream of input gene or genes. The corresponding regulatory elements and the display of detailed TF provide important information for molecular scientists studying individual genes.

4) PlantRegMap:<http://plantregmap.gao-lab.org/> Plant Transcriptional Regulatory Map. PlantRegMap provided a comprehensive, high-quality resource of plant transcription factors (TFs), regulatory elements and interactions between them, advancing the understanding of plant transcriptional regulatory system.

5) AGRIS-The Arabidopsis Gene Regulatory Information Serve: <http://arabidopsis.med.ohio-state.edu/>. You can query the name, sequence, location information and combined TF of TFBS on each gene of Arabidopsis.

6)) CIS-BP-online TFs and their binding motifs: <http://cisbp.cabr.utoronto.ca/>.

A comprehensive website about TF, providing TF information query and sequence analysis of TFBS and potential TF. The Catalog of Inferred Sequence Binding Preferences (CIS-BP) is a library of transcription factor (TF) DNA binding motifs and specificities. The data are organized in a user friendly manner for ease of searching, browsing, and downloading. CIS-BP also includes built-in web tools for scanning DNA sequences for putative TF binding sites, predicting the DNA binding motif of a given TF, and identifying a TF that might recognize a given DNA motif.

3. The author needs to prove how the database can help scientists explore the potential relevance of new biological information or data across multiple species?

4. The regulatory information the author mentioned in the manuscript are not available in the platform, including TF interaction regulatory network, promoters, enhancers, etc.

5. Most of the analysis results in the manuscript, including FFL, tissue-specific enhancers and

promoters, and the chromatin status of different species are only descriptive information and do not explain specific biological issues. In addition, these results are not reflected on the ChIP-Hub platform. What is the substantial relationship between these analyses and the platform?

5. In terms of website operation, first, it is not as friendly as the author said. No data download was provided from the table shown, and no further analysis was provided. In addition, interactive functions such as page loading are not smooth enough. But I have to admit that the website built by R Shiny is indeed very beautiful, giving people a refreshing feeling. Secondly, the related network functions between the transcription factors mentioned in the article (Figure 3C) and the dynamic functions of tissue-specific regulatory elements (Figure 4) are not presented on the website. On the contrary, the platform mainly provides descriptive information of epigenetic data or result information of basic program operations (such as target genes, genomic regions), which are preliminary and do not bring any substantial help to users. As the core of the article, ChIP-Hub cannot fully perform the functions mentioned in the article, just like a semi-finished website.

Minor

1. The boxplots and text are mixed in Supplementary Fig.12,14,15,and 16
2. Some analysis tools didn't work well such as signal tracks and signal plot visualization in ChIP-Hub.
3. The legend b and c of Figure1 are reversed.
4. The cluster boundary in Figure 5a is not clear, it's best to set gap to separate them.
5. About 30% of the documents were published ten years ago. There are too many old ones.

Reviewer #2:

Remarks to the Author:

In this manuscript, Fu and colleagues collected >10,000 datasets to build the ChIP-hub platform which is stunning and easy to follow. They have evaluated the quality of plant regulome and epigenome data and built the TF regulatory networks. Further, the authors have also identified many tissue specific promoters and enhancers and grouped them into ten clusters with kinds of function. Finally, each genome has been segmented into several states to study the correlation between chromatin states and regulator binding profile.

This work is of general interest to plant biologists because it provides a good platform to study TF and epigenetics. However, there are still some significant concerns which temper enthusiasm for this manuscript and should be addressed before it is acceptable.

Main comments:

The significant TF co-associations are defined as their co-association scores larger than the overall median value which means that half of them will be defined as significant co-associations. Such criteria will significantly increase the false discovery rate. Further, what would the network in Fig. 3b look like for a random set of TF co-associations, will these three modules also formed?

The chromatin stats identified in this work is mainly derived from the ChIP-seq data. While the DNA methylation level also plays a key role in the regulation of chromatin stats so that please provide more information that it is suitable to identify chromatin states only with several histone modification marks.

Minor comments:

Fig. 1b and Fig. 1c are labelled opposite. Authors call out "Fig. 1b" to refer to Fig. 1c

Fig. 3b, to help the readers better understand the TF co-association network, please provide more information about the co-association network (like width of edges represent for the co-association score and size of node for its degree).

Fig. 3e, 66.4% (377/568) of FFLs identified in previous study can also be identified in this work. The number of FFLs in this is 13,253 which much larger than 568 in previous study. Whether such overlap ratio (66.4%) are significant or even if some randomly mimic FFLs can also reached such ratio?

The order of manuscripts should be changed. Lines 165-174 talk about the sequence grammar and refer to Fig. 4e. Such paragraph should move to line 157 (before Fig 5) or the Fig 4e should rename to Fig 5c.

The association between target genes and regulatory elements based on the nearest neighbor strategy. However, the chromatin loops can help the formation of interactions between regulator and its target genes. In this regard, I am not sure if nearest neighbor strategy is suitable.

Reviewer #3:

Remarks to the Author:

The authors have manually curated a comprehensive collection of published plant chromatin/TF datasets and uniformly analyzed them. This includes a very large number of chromatin and TF ChIP-seq, DAP-seq, and open chromatin dataset across many plant species. Using this rich datasource the authors perform several interesting characterizations of plant chromatin and TF binding properties including describing tissue-specific chromatin dynamics, putative enhancers, and global chromatin states. They also developed a website to access this curated dataset and provide useful tools for exploring the data. Overall, the study includes some very insightful analysis of the transcription factor and chromatin landscape in plants and provides great resources for future studies.

Major issue:

The comprehensive, curated, and uniformly analyzed datasource of published ChIP-seq, DAP-seq, and ATAC-seq datasets is an extremely valuable resource and can be leverage for a wide-range of analysis. The authors' website Chip-Hub is an excellent tool to interface with this database and I believe many researchers will use it. However, I am always concerned that a paper-associated website may not stay active and available in perpetuity. By adding a few data tables with key information as supplemental tables and/or extended data to this paper would ensure it will be available in the future. Specifically, the authors could include one table to capture the metadata for all study samples (Paper citation, SRX/DRX/SRA number, ChIP/DAP TF or chromatin mark etc). Additionally, narrow and broad peak files in BED format for all factors would be incredibly valuable.

Once again the authors website and its tools are quite useful but including a table of the metadata for the literature datasets (study citation, SRX etc) and BED files of the narrow and broad peaks calls for all experiments would ensure they will always be available. I realize the file size may be large but as they are all text tiles even 10K experiments in zipped format should be a manageable size.

Minor issue:

Several figures may be improved through some reduction in the complexity to make them more readable. There are so many tracks included that its difficult to read individual labels as font sizes are very small. For example, Figure 4 and 5 have a lot of genome tracks and it would be possible to remove some replicate tracks that have very similar information and still capture the overall trends. In Figure 5b the browser track has multiple replicates for each tissue (12 root, 4 root hair etc) but each within group (i.e. all 12 root tracks) have very similar information. It seems like 2-3 representatives from each would be sufficient to see global tissue pattern. This would allow font to increase and make

the overall figure easier to read. The original large figure could be put in the supplement if you still want to show it. For Figure 4a you could remove everything but M1-M3 as few other TFs outside of these groups do not show strong correlations. Figure 4b could also then be reduced to reflect only that set of TFs. Once again, the full Fig 4a could be pushed into the supplement and a condensed and simplified version could be used in the main figure panel. In general, I think stylistically all the figures would be improved through some reduction in the complexity to make them more readable and easier to digest.

REVIEWER COMMENTS

Response to all reviewers:

We are very grateful to the reviewers for their positive and constructive comments on our manuscript. We believe that their valuable suggestions have helped us to improve the manuscript a lot. We have updated the manuscript accordingly and provide below point-by-point responses to the reviewers' comments. Note that changes in manuscript are highlighted in **blue**.

Reviewer #1 (Remarks to the Author):

Fu et al. introduced a website ChIP-Hub, used to summarize and display ChIP-seq data in plants. However, many platforms have released similar data sets, providing richer tools for data mining, while the function of ChIP-Hub is rather limited. I did not discover the uniqueness of the platform, nor did it elaborate on specific biological issues. It is just another form of accumulation of existing data and a description of corresponding information. Even if this information is sorted out in great detail, it is of little help for functional research. Nevertheless, I have to admit that the article layout and website design are very good.

Response: We thank the reviewer for his/her positive comments on our manuscript.

Major

1. The only feature of ChIP-Hub is the large amount of data, without in-depth analysis. All data comes from public databases and can be processed within 3-4 weeks without any technical difficulties. Apart from storing public data, I am not sure what the purpose of this platform is.

Response: We appreciate the reviewer's positive as well as critical comments on our manuscript. We believe that a main strength of our manuscript lies in extensive and careful data curation based on information from publications and/or raw description from databases. It's a very time-consuming step in our work. We described this in details at the Section of "Data source, curation and collection" in the manuscript. On one hand, unlike data from the human/mouse ENCODE project, where all the metadata (i.e., replicates, experimental group -- ChIP or input) are clearly defined by data providers, a lot of plant ChIP-seq data storing in public databases without clearly formatted metadata, so it's impossible to process them in an automatic way. On the other hand, comparison of data from different studies and across plant species is not straightforward when data were not evaluated and processed in an uniform way. We believe these are the main obstacles of re-use of public ChIP-seq data for the plant community,

although there are already huge data contributed by the community so far. Given this background, we established the ChIP-Hub platform from 2015 with an aim of uniform reanalysis and comprehensive evaluation of plant regulome data.

2. There are several representative plant regulatory element websites, all with specific functions. There are many functional overlaps. Where is the innovation of ChIP-Hub ?

1) PlantCistromeDB-Atlases of the Plant Cistrome and Epicistrome: http://neomorph.salk.edu/dap_web/pages/index.php , which contains TFBS information, and you can conveniently view the combination of different TFs on the genome at the same time on the browser Site.

2) PlantTFDB: <http://planttfdb.gao-lab.org/> , which mainly collects comprehensive information on transcription factors in plants, and provides a convenient entry for researchers to understand the information of transcription factors of interest. It can also predict the transcription factor of a specific gene or protein sequence.

3) Plant regulomics: <http://bioinfo.cemps.ac.cn/plant-regulomics/> , which helps retrieve regulatory elements up and downstream of input gene or genes. The corresponding regulatory elements and the display of detailed TF provide important information for molecular scientists studying individual genes.

4) PlantRegMap:<http://plantregmap.gao-lab.org/> Plant Transcriptional Regulatory Map. PlantRegMap provided a comprehensive, high-quality resource of plant transcription factors (TFs), regulatory elements and interactions between them, advancing the understanding of plant transcriptional regulatory system.

5) AGRIS-The Arabidopsis Gene Regulatory Information Serve: <http://arabidopsis.med.ohio-state.edu/>. You can query the name, sequence, location information and combined TF of TFBS on each gene of Arabidopsis.

6)) CIS-BP-online TFs and their binding motifs: <http://cisbp.cabr.utoronto.ca/>.

A comprehensive website about TF, providing TF information query and sequence analysis of TFBS and potential TF. The Catalog of Inferred Sequence Binding Preferences (CIS-BP) is a library of transcription factor (TF) DNA binding motifs and specificities. The data are organized in a user friendly manner for ease of searching, browsing, and downloading. CIS-BP also includes built-in web tools for scanning DNA sequences for putative TF binding sites, predicting the DNA binding motif of a given TF, and identifying a TF that might recognize a given DNA motif.

Response: We appreciate the reviewer's constructive comments. These databases are indeed useful and valuable for studies of plant regulatory elements by providing specific functions. Our ChIP-Hub database is different from these databases in terms of data amount and function specificity. On the one hand, as the reviewer recognized, ChIP-Hub contains the most comprehensive regulomic datasets in plants, with at least ten times of data in "Plant regulomics" or PlantRegMap. PlantCistromeDB contains DAP-seq

datasets almost in *Arabidopsis thaliana*, which were generated by a specific study. CIS-BP provides comprehensive TFBS information based on DNA motifs across various species. However, TFBSs predicted by omics data are more useful in terms of accuracy and biological context relevance. Nevertheless, most of the data information from these databases mentioned by the reviewer has been included in our ChIP-Hub database including TF annotation from PlantTFDB and DNA motifs from CIS-BP and JASPAR (all datasets collected by “Plant regulomics”, PlantRegMap and PlantCistromeDB have also been included in ChIP-Hub). Beside this, we also evaluated the quality of each datasets using the ENCODE standard so that users can choose proper datasets in their analyses. On the other hand, ChIP-Hub allows for comparative genomics analyses across plant species (as demonstrated in **Figures 6 and 7**), providing a unique feature among these databases. We have added the following discussion in the revision: ‘The ChIP-Hub platform is different from several representative plant regulatory element databases, including Plant Cistrome Database, ReMap, Plant Regulomics, PlantRegMap, AGRIS, JASPAR and CIS-BP, in terms of data content, data amount and function specificity. Omics-related datasets collected in ChIP-Hub far exceed data in all those relevant databases. In addition, ChIP-Hub provides predicted TF binding site (TFBS) information using DNA motifs taken from CIS-BP and JASPAR. Most importantly, ChIP-Hub allows comparative regulomic analyses, which provides a unique feature of ChIP-Hub among similar databases.’

3. The author needs to prove how the database can help scientists explore the potential relevance of new biological information or data across multiple species?

Response: We thank the reviewer for bringing up critical questions and constructive criticisms. In our previous version of manuscript, we actually performed several related analyses to explore new biological information from such large data, including identification of TF co-associations and potential regulatory network loops (**Fig. 3**), prediction of tissue-specific regulatory elements (active promoters and enhancers) and underlying sequence grammar (**Figs. 4 and 5**), and comparative analysis of chromatin states (**Fig. 7** in revision). Such comprehensive analyses have not been performed in plants so far mostly due to lack of such well-processed big data. In order to further prove the power of ChIP-Hub in comparative genomics analysis, we conducted additional analysis in the revised manuscript. Specifically, we are interested in tracking the evolution of active functional elements (including promoters and enhancers) across 17 plant species (**Fig. 6** in the revision, see below), using a similar strategy by Villar et al. (*Cell* 2015, doi:10.1016/j.cell.2015.01.006) in mammalian species. We identified a list of promoters and enhancers with different degree of conservation and enriched with specific biological functions. We found that promoter activity is evolutionarily stable while enhancer evolution is relatively rapid, reflecting a fundamental characteristic of the regulatory genome in eukaryote.

Fig. 6. Evolutionarily tracking plant promoters and enhancers. **(a)** Phylogenetic tree showing the evolutionary relationships of plant species used in the analysis, including five monocots and twelve dicots. **(b)** The number

of predicted promoters and enhancers in each species. (c) Distance of peak summits to the transcription start site (TSS). (d) Sankey plot showing conserved regulatory elements among seven representative species, using Arabidopsis as a reference. Each line refers active regulatory element (promoter or enhancer) is alignable to the Arabidopsis genome. (e) Dotplots showing the number of species in which the Arabidopsis promoter (above) or enhancer (below) is alignable. Top conserved promoters and enhancers are labeled and four examples are highlighted in (g). (f) Barchart summarizing the degree of conservation of promoters and enhancers in each species. (g) Shown are examples of regulatory regions active in different plant species. (h) Enrichment analysis of gene ontology (GO) biological pathways for promoters and enhancers with different degree of conservation.

4. The regulatory information the author mentioned in the manuscript are not available in the platform, including TF interaction regulatory network, promoters, enhancers, etc.

Response: We have now included these analyzed data in the ChIP-Hub website (see screenshots below). The data can also be downloaded from the website for user purpose analysis.

Specific regular type
 Enhancer Promoter

Copy CSV Excel PDF Print

Select: All Type: All Coordinate: All Target Gene: All Annotation: All

Select	Type	Coordinate	Target Gene	Annotation
[ ]	Enhancer	Chr1:74-729	All tissues	Intergenic
[ ]	Enhancer	Chr1:1250-1843	All tissues	Intergenic
[ ]	Promoter	Chr1:1957-3961	All tissues	Intergenic
[ ]	Promoter	Chr1:4483-5044	All tissues	Exon
[ ]	Enhancer	Chr1:5304-5401	All tissues	Intron
[ ]	Enhancer	Chr1:6447-6803	All tissues	3'UTR
[ ]	Promoter	Chr1:8087-9108	All tissues	Intergenic
[ ]	Promoter	Chr1:9475-9834	All tissues	Intergenic
[ ]	Enhancer	Chr1:10888-11543	All tissues	Intergenic
[x]	Promoter	Chr1:11905-12963	All tissues	Exon

Showing 1 to 10 of 75,739 entries

Select one row in the table above to visualize associated peak track

Overview of sequences correlation of peaks

Minimum Number of Peaks: 1000

Copy CSV Excel PDF Print

Action	Select	Sample	Tissue	Peak Statistic	Treatment	SPOF	FRIP	URA Project	Classification
[ ] Like it [ ] Download	[ ]	SRX7780580	testis	Enhancer Number: 23647 Promoter Number: 18331	Treatment Drought Description Drought_in_the_Whetman_paper_no_3_Nr_1_3	0.454	0.51371	SRP230345	FAIRE-seq
[ ] Like it [ ] Download	[x]	SRX7780581	testis	Peak Num: 39718 Enhancer Number: 20843 Promoter Number: 18332	Treatment Drought Description Drought_in_the_Whetman_paper_no_3_Nr_1_3	0.4642	0.49374	SRP230345	FAIRE-seq
[ ] Like it [ ] Download	[ ]	DRX123174	testis	Peak Num: 4343 Enhancer Number: 2149 Promoter Number: 4301	Treatment DMSO Description DMSO_treatment_for_2_days	0.8801	0.19513	DRP004220	ATAC-seq

Showing 1 to 10 of 448 entries

Select one row in the table above to download associated Enhancer and Promoter for selected sample.

Visualization in EpiBrowser

Track color order by: Default Tissue

Track color style: Uniform Random ColorMap

ColorMap: Set3

Track height: 38

Step1. Choose peak position to visualize

Step2. Choose relevant datasets

Step3. Click 'Visualize' button to show peak tracks

Screenshot of gene regulatory networks on the ChIP-Hub website.

ChIP-Hub Home Browser Tools Statistics Download Help -

Arabidopsis thaliana Nothing selected

Hint: the tables in each tab below can be filtered by typing your keywords in the search or input box. Click rows for selection or deselection. Multiple rows could be selected.

Samples Experiments Genes CREs Networks

Show 10 entries Search: Show 10 entries Search:

ID	TFName	ID	GeneName
AT3G07610	IBM1	AT1G01725	AT1G01725
AT4G16310	LDL3	AT1G09910	AT1G09910
AT3G10390	FLD	AT1G10610	AT1G10610
AT1G24260	AGL9	AT1G13230	AT1G13230
AT5G18960	FRS12	AT1G20270	AT1G20270
AT4G14720	PPD2	AT1G20280	AT1G20280
AT1G49950	ATTRB1	AT1G22540	AT1G22540
AT3G58190	ASL16	AT1G25240	AT1G25240
AT2G33860	ARF3	AT1G25230	AT1G25230
AT5G42520	ATBPC6	AT1G26100	AT1G26100

Showing 1 to 10 of 157 entries Previous 1 2 3 4 5 ... 16 Next Clear Search

Select rows in the table above to choose TFs and regulated Genes visualize associated network.

Show Names Show name or not

Step1. Choose TFs Step2. Choose Genes Step3. Click 'Draw Network' button

Screenshot of the predicted promoters and enhancers on the ChIP-Hub website.

5. Most of the analysis results in the manuscript, including FFL, tissue-specific enhancers and promoters, and the chromatin status of different species are only descriptive information and do not explain specific biological issues. In addition, these results are not reflected on the ChIP-Hub platform. What is the substantial relationship between these analyses and the platform?

Response: We thank the reviewer for the valuable comments. We have addressed the reviewer's concerns by adding more explanations about the biological implications of our analysis results, as outlined below:

- 'Furthermore, we validated the confidence of predicted FFLs using known gene interactions from the flowering pathways (Bouché et al. *Nucleic Acids Res.* 44, D1167–D1171 (2016)). For example, regulatory loops involved in the flowering-time regulation (Amasino R. *Plant J.* 61(6):1001-13. (2010)) and the antagonistic interaction between class A and class C genes in the ABCE model of flower development (Grigороva et al., *Development* 138, (2011)) have been

confirmed by our data (**Fig. 3f**). In sum, the above analysis provides a rich resource to study the biological role of regulatory loops in specific contexts.’

- ‘Regulatory elements in clusters 2 (C2) and C3 are highly active in flower-related tissues, and their target genes largely involved in biological processes such as ‘flower development’ and ‘floral organ development’ (**Supplementary Table 6**), including a list of well-known genes controlling floral transition and flower development (Bouché et al. *Nucleic Acids Res.* 44, D1167–D1171 (2016)), such as *LEAFY (LFY)*, *APETALA1 (AP1)*, *FRUITFULL (FUL)*, *STERILE APETALA (SAP)* and *AGAMOUS-LIKE 24 (AGL24)*. Regulatory elements in C4 and C7 are specifically active in root- and leaf-related tissues, with target genes in ‘response to biotic stimulus’ and ‘defense response’, respectively. For example, *NAC1* has shown to mediate auxin signaling to promote lateral root development (Xie et al. *Genes Dev.* 14, (2000)), while *YY1* is an important regulator of the ABA response network for plant growth and development (Li et al. *Mol. Plant* 9, (2016)).’
- ‘... The above findings indicate that plants may share a conserved histone code for gene regulation.’

FFLs are available in **Supplementary Table 3**. Tissue-specific enhancers and promoters are now available in the ChIP-Hub website under the “CREs” tab of the “Browser” panel. Results of chromatin states can also be visualized and downloaded from the website.

In our previous design, the ChIP-Hub platform generally contained result data from the pipeline but without in-depth analysis. The analysis results in the manuscript were only performed with intended focus in specific species and would be supplied with the manuscript. The main idea for this design was that users usually have different analysis plans and the general data files from the pipeline would be enough

for their analysis. In the revision, we took the reviewer's nice suggestion and showed the analysis results in the ChIP-Hub website as well. We stated this in the manuscript.

5. In terms of website operation, first, it is not as friendly as the author said. No data download was provided from the table shown, and no further analysis was provided. In addition, interactive functions such as page loading are not smooth enough. But I have to admit that the website built by R Shiny is indeed very beautiful, giving people a refreshing feeling. Secondly, the related network functions between the transcription factors mentioned in the article (Figure 3C) and the dynamic functions of tissue-specific regulatory elements (Figure 4) are not presented on the website. On the contrary, the platform mainly provides descriptive information of epigenetic data or result information of basic program operations (such as target genes, genomic regions), which are preliminary and do not bring any substantial help to users. As the core of the article, ChIP-Hub cannot fully perform the functions mentioned in the article, just like a semi-finished website.

Response: We thank the reviewer's positive comments and constructive criticisms. We acknowledge the website operation may not be friendly (e.g., dis-connection problem) and smooth enough (i.e., slow loading at the first time) in some way as it's developed based on the Shiny Application – the FREE version of Shiny Server only executes the operation for a single user. However, we have the plan to develop a new version ChIP-Hub to improve these issues in the near future. Regarding to the issue about data download, we implemented functions in the platform for downloading single dataset just by clicking a button. For batch download, we now provided a separated link for users to get data of interest.

We have revised the issue about visualization of gene regulatory networks and tissue-specific regulatory elements in the website. Please also refer to our above **Response #5**.

Minor

1. The boxplots and text are mixed in Supplementary Fig.12,14,15,and 16

Response: Thank you to point out this. We have fixed the problem in the revision (these figures are now numbered to **Supplementary Fig.15,17,18 and 19**).

2. Some analysis tools didn't work well such as signal tracks and signal plot visualization in ChIP-Hub.

Response: This is an aware issue as ChIP-Hub employs deepTools (<https://deeptools.readthedocs.io>) to generate signal tracks and signal plot visualization. It would be a bit more patient to run this analysis tool in ChIP-Hub. However, for batch analysis, we would recommend users to download signal and peak files

to run the job locally. To this end, we have provided example codes so that users can run such analysis by themselves in case of batch analysis.

3. The legend b and c of Figure 1 are reversed.

Response: Thank you to point this out. We have fixed this issue.

4. The cluster boundary in Figure 5a is not clear, it's best to set gap to separate them.

Response: We have revised the **Figure 5a** according to the reviewer's suggestion.

5. About 30% of the documents were published ten years ago. There are too many old ones.

Response: Thank you for this comment. We have updated all relevant online documents. Please check this at our ChIP-Hub website (<https://biobigdata.nju.edu.cn/ChIPHub/>).

Reviewer #2 (Remarks to the Author):

In this manuscript, Fu and colleagues collected >10,000 datasets to build the ChIP-hub platform which is stunning and easy to follow. They have evaluated the quality of plant regulome and epigenome data and built the TF regulatory networks. Further, the authors have also identified many tissue specific promoters and enhancers and grouped them into ten clusters with kinds of function. Finally, each genome has been segmented into several states to study the correlation between chromatin states and regulator binding profile.

This work is of general interest to plant biologists because it provides a good platform to study TF and epigenetics. However, there are still some significant concerns which temper enthusiasm for this manuscript and should be addressed before it is acceptable.

Response: We thank the reviewer for his/her positive comments on our manuscript.

Main comments:

The significant TF co-associations are defined as their co-association scores larger than the overall median value which means that half of them will be defined as significant co-associations. Such criteria will significantly increase the false discovery rate. Further, what would the network in Fig. 3b look like for a random set of TF co-associations, will these three modules also formed?

Response: There was a mistake in our previous description and we apologize for this. The threshold was actually set as 0.2 in our analysis, which was determined by an elbow statistic (see a new Figure in

Supplementary Fig. 9a as shown below). We have corrected this in the revised manuscript: “Significant TF co-associations are defined as their co-association scores larger than 0.2, an optimal threshold determined by an elbow statistic (**Supplementary Fig. 9a**).”

To answer the second comment from the reviewer, we generated a random set of TF co-associations using the exactly same co-association scores. We visualized the TF co-associations into a network in a similar way. However, we found no isolated module from such randomization analysis (see **Supplementary Fig. 9b** as shown below; we repeated the analysis for >10 times and got similar results).

Supplementary Fig. 9. Co-associations of TFs, related to Fig. 3. (a) Determining an optimal threshold (highlighted in red) of significant TF co-associations based on an elbow statistic. (b) Network showing a random co-associations between TFs. Co-association scores follow the same distribution as observed. However, no significant co-association modules were observed.

The chromatin states identified in this work is mainly derived from the ChIP-seq data. While the DNA methylation level also plays a key role in the regulation of chromatin states so that please provide more information that it is suitable to identify chromatin states only with several histone modification marks.

Response: We thank the reviewer for this very nice suggestion. We agree with the reviewer that DNA methylation data would be useful for identification of chromatin states. Indeed, such investigation has recently been made in plants by several studies, e.g., Zhao et al., *Nat Commun* (2020), doi: <https://doi.org/10.1038/s41467-020-16457-5> and Liu et al., *Nucleic Acids Research* (2018), doi: <https://doi.org/10.1093/nar/gkx919>. Since the main goal of this analysis is to show the power of ChIP-Hub for comparative analysis of the chromatin states across species, and since DNA methylation data are not included in our ChIP-Hub platform so far, we would rather like to investigate the issue raised by the reviewer in some independent analysis in the future.

Minor comments:

Fig. 1b and Fig. 1c are labelled opposite. Authors call out “Fig. 1b” to refer to Fig. 1c

Response: We appreciate the reviewer to point this out. We have fixed this issue in the revision.

Fig. 3b, to help the readers better understand the TF co-association network, please provide more information about the co-association network (like width of edges represent for the co-association score and size of node for its degree).

Response: Thank you very much for this nice suggestion. In the revision, we have revised the Fig. 3b and clarified this in the figure legend by adding the following information: “The width of edge represents for the co-association score and the size of node for its degree”.

Fig. 3b. Network showing significant co-associations between TFs. Significant TF co-associations are defined as their co-association scores larger than 0.2, an optimal threshold determined by an elbow statistic (Supplementary Fig. 8c). Three highly interplayed modules are highlighted. The width of edge represents for the co-association score and the size of node for its degree.

Fig. 3e, 66.4% (377/568) of FFLs identified in previous study can also be identified in this work. The number of FFLs in this is 13,253 which is much larger than 568 in previous study. Whether such overlap ratio (66.4%) are significant or even if some randomly mimic FFLs can also reach such ratio?

Response: We have tested the significance of the overlap ratio by χ^2 test. The overlap ratio is indeed significant (p -value $< 2.2e-16$).

The order of manuscripts should be changed. Lines 165-174 talk about the sequence grammar and refer to Fig. 4e. Such paragraph should move to line 157 (before Fig 5) or the Fig 4e should rename to Fig 5c.

Response: Thank you for this comment. We have moved the paragraph (lines 165-174) to line 157.

The association between target genes and regulatory elements based on the nearest neighbor strategy. However, the chromatin loops can help the formation of interactions between regulator and its target genes. In this regard, I am not sure if nearest neighbor strategy is suitable.

Response: We appreciate the reviewer's comments. The nearest neighbor strategy might lead to false target associations due to chromatin loops. We discussed this issue by adding the following sentences: "Note that the nearest neighbor strategy could lead to false target associations due to chromatin loops which can help the formation of interactions between regulator and its target genes. Nevertheless, the chromatin conformation in Arabidopsis is dominantly represented by kb-sized interactive regions based on Hi-C analyses (Wang et al., *Genome Res.* 2005, 25, 246-256; Feng et al., *Mol. Cell* 2014, 55, 694-707), which indicates that enhancers mostly target their neighboring gene(s) in Arabidopsis."

Reviewer #3 (Remarks to the Author):

The authors have manually curated a comprehensive collection of published plant chromatin/TF datasets and uniformly analyzed them. This includes a very large number of chromatin and TF ChIP-seq, DAP-seq, and open chromatin dataset across many plant species. Using this rich datasource the authors perform several interesting characterizations of plant chromatin and TF binding properties including describing tissue-specific chromatin dynamics, putative enhancers, and global chromatin states. They also developed a website to access this curated dataset and provide useful tools for exploring the data. Overall, the study includes some very insightful analysis of the transcription factor and chromatin landscape in plants and provides great resources for future studies.

Response: We thank the reviewer for his/her positive comments on our manuscript.

Major issue:

The comprehensive, curated, and uniformly analyzed datasource of published ChIP-seq, DAP-seq, and ATAC-seq datasets is an extremely valuable resource and can be leverage for a wide-range of analysis. The authors' website Chip-Hub is an excellent tool to interface with this database and I believe many researchers will use it. However, I am always concerned that a paper-associated website may not stay active and available in perpetuity. By adding a few data tables with key information as supplemental tables and/or extended data to this paper would ensure it will be available in the future. Specifically, the authors could include one table to capture the metadata for all study samples (Paper citation, SRX/DRX/SRA number, ChIP/DAP TF or chromatin mark etc). Additionally, narrow and broad peak files in BED format for all factors would be incredibly valuable.

Response: We thank the reviewer's positive comments on our manuscript. To address the reviewer's concerns, we have provided a metadata file for all study experiments as well peak files in BED format and deposited them at the Zenodo database (with DOI: <https://doi.org/10.5281/zenodo.5912234>).

Once again the authors website and its tools are quite useful but including a table of the metadata for the literature datasets (study citation, SRX etc) and BED files of the narrow and broad peaks calls for all experiments would ensure they will always be available. I realize the file size may be large but as they are all text files even 10K experiments in zipped format should be a manageable size.

Response: We thank the reviewer again for the comments. We have done this according to the reviewer's suggestions. See our response above.

Minor issue:

Several figures may be improved through some reduction in the complexity to make them more readable. There are so many tracks included that its difficult to read individual labels as font sizes are very small. For example, Figure 4 and 5 have a lot of genome tracks and it would be possible to remove some replicate tracks that have very similar information and still capture the overall trends. In Figure 5b the browser track has multiple replicates for each tissue (12 root, 4 root hair etc) but each within group (i.e. all 12 root tracks) have very similar information. It seems like 2-3 representatives from each would be sufficient to see global tissue pattern. This would allow font to increase and make the overall figure easier to read. The original large figure could be put in the supplement if you still want to show it. For Figure 4a you could remove everything but M1-M3 as few other TFs outside of these groups do not show strong correlations. Figure 4b could also then be reduced to reflect only that set of TFs. Once again, the full Fig 4a could be pushed into the supplement and a condensed and simplified version could used in the main figure panel. In general, I think stylistically all the figures would be improved through some reduction in the complexity to make them more readable and easier to digest.

Response: We thank the reviewer again for the suggestions. We revised both figures (**Figures 3 and 4** in revision) as the reviewer suggested. For **Fig. 3a**, we only showed co-associations among TFs in M1-M3 in the main Figure and put the original large figure in **Supplementary Figure 8**. For **Fig. 3b**, only ChIP-seq signal tracks for TFs in M1 are shown. For **Fig. 4b**, we chose 2 representative replicates for each tissues to show in the main Figure and put the original large figure in the supplement (**Supplementary Figure 11**).

Reviewers' Comments:

Reviewer #1:

Remarks to the Author:

Nature Communication publishes "novel and important research study of high quality" as indicated in the homepage. However, this manuscript is far from novel and important.

1. It lacks biological significance, and some major biological analyses that have been shown in the figure do not exist on the analysis page of the website.

Despite the revision touched biological issues as shown in Fig 6, the evolution function shown in Figure 6 almost has no corresponding analysis page on the website, and the related analysis function of Chromatin state is not implemented on the website. The website is like a vase with an empty interior. Almost no tool providing biologically meaningful analysis, or to further leverage the data to produce biologically meaningful results. All are brief introductions and statistics of the data, which can be easily obtained from the original GEO website.

2. Data update and maintenance is a big problem. Some of the current reference genome versions are already old.

The author emphasized that the advantage of the ChIP-Hub database is the increase in the amount of data, and the author also showed that the data processing process of the ChIP-Hub database is a very time-consuming task (data collection and processing began in 2015, and it took 7 years to be processed only once). With the rapid change of genome version and gene annotation, whether the website can match the new genome version in time, especially the target genes and regions that may change. The long time to process the data reflects the unavailability of timely data update. When there is a new release of the genome or annotation, each data need to be re-processed, from alignment to visualization file generation. Accordingly, some of the reference genome and gene model version are already outdated. How to solve the issue of data update is the main problem of the website.

3. No option for parameter adjustment and personalized analysis.

The peak results including .bed and .bw files shared on the platform are the results of a single parameter. In most cases, these results cannot be used directly, and the parameters need to be adjusted according to specific situations.

3. Inaccuracy. The target genes of regulatory elements such as enhancers and promoters predicted by the platform are completely determined according to the principle of proximity. Conceptually, enhancers are not limited by the distance to target genes. Enhancers generally pull the physical distance from the target gene through the folding of chromatin in three-dimensional space to achieve long-range regulation of target genes. Therefore, it is unreasonable to assign enhancer target merely based on proximity. This situation requires the support of more types of data such as Hi-C data, and this platform lacks the collection of such data.

4. Inconvenience: The potential users of the platform can be roughly divided into two types: one is to study the molecular mechanism of specific genes, and the other is to do big data analysis. The platform is not attractive enough for either type.

5. For visitors who study specific molecular mechanisms, their purpose is to quickly obtain information such as the regulatory relationship of the gene(s) of interest. This platform didn't provide such services. The starting point of the analysis of this platform is based on the collected public data sets. There is no unified classification and integration of these data sets in advance, resulting in the lack of prominent functions and not attractive to users interested in biological mechanism.

6. For users who do big data analysis, the analysis often involves multiple data sets. The upper limit of a single analysis of this platform is only 10 data sets, which is inefficient and cannot meet the purpose of such users.

8. Instability. I have tried several analyses on this platform, but most of the time, I failed, which directly leads to disconnection from the server, making the user experience extremely poor.

Reviewer #2:

Remarks to the Author:

Thank Fu and colleagues for the response. It is very important in this era of omics to collect and re-analyze the omics data. In this manuscripts, Fu, et al collected > 10,000 datasets from > 40 plant species building a website that provides huge resource for future research. In this manuscript, some insight analysis is very useful for studying the relationship of TF and histone modifications like TF interaction regulatory network and tissue-specific regulatory elements etc. One of the most important research points for plants is how plants adapt to changing environments. If this part of data can be collected and analyzed in this research, it will make the paper more outstanding.

Reviewer #3:

Remarks to the Author:

The changes and additions the authors made to the manuscript have satisfied my previous concerns and requests.

Reviewer #1 (Remarks to the Author):

Nature Communication publishes “novel and important research study of high quality” as indicated in the homepage. However, this manuscript is far from novel and important.

1. It lacks biological significance, and some major biological analyses that have been shown in the figure do not exist on the analysis page of the website.

Despite the revision touched biological issues as shown in Fig 6, the evolution function shown in Figure 6 almost has no corresponding analysis page on the website, and the related analysis function of Chromatin state is not implemented on the website. The website is like a vase with an empty interior. Almost no tool providing biologically meaningful analysis, or to further leverage the data to produce biologically meaningful results. All are brief introductions and statistics of the data, which can be easily obtained from the original GEO website.

> **Response:** We thank the reviewer to point this out. The reviewer said our work “lacks biological significance” but did not provide any detailed comments. However, two other referees recognized the importance of our work. Both agree that “... *insight analysis is very useful for studying the relationship of TF and histone modifications like TF interaction regulatory network and tissue-specific regulatory elements etc. ...*” and “*the study includes some very insightful analysis of the transcription factor and chromatin landscape in plants and provides great resources for future studies.*”

Regarding the data issue, we indeed have provided both the uniformly processed data files and reanalysis results with specific biological points (Figures 3-7). The uniformly processed data would be useful for users who would like to check specific dataset(s). These data have been presented in the website. The reanalysis results such as chromatin states were obtained based on some selected datasets (not the whole datasets) in ChIP-Hub. These data files are available for download and can now be visualized in the ChIP-Hub genome browser. Conserved enhancers and promoters are also provided for download.

2. Data update and maintenance is a big problem. Some of the current reference genome versions are already old.

The author emphasized that the advantage of the ChIP-Hub database is the increase in the amount of data, and the author also showed that the data processing process of the ChIP-Hub database is a very time-consuming task (data collection and processing began in 2015, and it took 7 years to be processed only once). With the rapid change of genome version and gene annotation, whether the website can match the new genome version in time, especially the target genes and regions that may change. The long time to process the data reflects the unavailability of timely data update. When there is a new release of the genome or annotation, each data need to be re-processed, from alignment to visualization file generation. Accordingly, some of the reference genome and gene model version are already outdated. How to solve the issue of data update is the main problem of the website.

> **Response:** We thank the reviewer for these critical comments. Firstly, we have to point out that reference genome versions do not change so fast in plants. We have confirmed that the reference genomes used in our analysis are still the widely-used versions by the research community or in publications. For example, the Arabidopsis and rice reference genomes remain the most recently updated versions. Other less used/non-model plant reference genomes are even not changed at all since their first release. Secondly, data update for a new gene model version is easy to work from peak files. Lastly, the reviewer has to acknowledge that standardized genomic data collection, uniform processing and reuse remain a big issue in plant research but there is limited such attempt so far. It's very labor-expensive and time-consuming and generally requires consortium effort. Like the update of a genome version, data update in ChIP-Hub also requires extensive work and more time. But we believe it's doable either based on genome coordinate transfer by 'liftOver' or re-analyze the data based on a new genome version. In short, we strongly believe that the currently used genome versions by ChIP-Hub can be useful for users in most cases.

3. No option for parameter adjustment and personalized analysis.

The peak results including .bed and .bw files shared on the platform are the results of a single parameter. In most cases, these results cannot be used directly, and the parameters need to be adjusted according to specific situations.

> **Response:** The reviewer appears to have misunderstood this point. As we have clearly stated in the Methods and also shown in the websites: we have provided several peak sets with different statistical thresholds for personalized analysis. The corresponding sentences are: “ ... ‘reproducible’ peaks across pseudo-replicates and true replicates with an IDR < 0.05 were recommend for analysis. Besides, peaks with different statistical thresholds are available upon user request. For example, ‘significant’ peaks were defined as a fold-change (fold enrichment above background) > 2 and a $-\log_{10}$ (q-value) > 3; while ‘lenient’ peaks as a fold-change > 2 and a $-\log_{10}$ (q-value) > 2. ‘Relaxed’ peaks without additional thresholding were also provided so that any custom threshold can be applied.”

3. Inaccuracy. The target genes of regulatory elements such as enhancers and promoters predicted by the platform are completely determined according to the principle of proximity. Conceptually, enhancers are not limited by the distance to target genes. Enhancers generally pull the physical distance from the target gene through the folding of chromatin in three-dimensional space to achieve long-range regulation of target genes. Therefore, it is unreasonable to assign enhancer target merely based on proximity. This situation requires the support of more types of data such as Hi-C data, and this platform lacks the collection of such data.

> **Response:** We thank the reviewer to raise this point. Target gene prediction for enhancers is still challenging in plants as there is no systemic evaluation so far. We do agree with the reviewer that Hi-C or ChIA-PET data may help for enhancer target prediction. The nearest neighbor strategy might lead to false target associations due to chromatin loops. We have discussed this issue in the revised version: “Note that the nearest neighbor strategy could lead to false target associations due to chromatin loops which can help the formation of interactions between regulator and its target genes. Nevertheless, the chromatin conformation in Arabidopsis is dominantly represented by kb-sized interactive regions based on Hi-C analyses (Wang et al., Genome Res. 2005,

25, 246-256; Feng et al., *Mol. Cell* 2014, 55, 694-707), which indicates that enhancers mostly target their neighboring gene(s) in *Arabidopsis*.” Besides, there are not always matched Hi-C datasets in specific plant species or tissue types so far. In some cases, Hi-C data are generally more noise (especially when the resolution is low) which would make target prediction even more inaccurate based on chromatin interactions. To make the results more comparable in ChIP-Hub, we thus adopted the principle of proximity for enhancer-target association. We will consider to collect such data for enhancer target prediction in ChIP-Hub in the future.

4. Inconvenience: The potential users of the platform can be roughly divided into two types: one is to study the molecular mechanism of specific genes, and the other is to do big data analysis. The platform is not attractive enough for either type.

> **Response:** The result files including peak and signal data can be loaded into the embedded Epigenome Browser and molecular biologists can find their interested genes either via the browser or the built-in search function for data visualization purpose (see a screenshot below for example). For people who do big data analysis, they can download the processed data in batch either from the ChIP-Hub download page (https://biobigdata.nju.edu.cn/ChIPHub_download/) or from the Zenodo repository (<https://doi.org/10.5281/zenodo.5912234>).

ChIP-Atlas - Home - Browse - Tools - Statistics - Download - Help

ChIP-Atlas

Hint: The tables in each tab below can be filtered by typing your keywords in the search or input box. Click on the selection or production. Multiple rows could be selected.

Specify gene sets:

 All genes

 mRNAs

 lncRNAs

 Protein-coding

Select gene-related signals
Choose the gene of interest

Action	Select	Coordinates	Gene ID	Gene Name	Transcripts	Description
View	[ ]	chr1:1,000,000-1,000,000	LOC101927295	CDK2	CDK2 (transcript)	F-box domain protein/CDK2 - F-box domain containing protein, cytoskeletal
View	[ ]	chr1:1,000,000-1,000,000	LOC101927295	CDK2	CDK2 (transcript)	imprinted protein
View	[ ]	chr1:1,000,000-1,000,000	LOC101927295	CDK2	CDK2 (transcript)	Cytosolic protein/CDK2 - cytoskeletal
View	[ ]	chr1:1,000,000-1,000,000	LOC101927295	CDK2	CDK2 (transcript)	imprinted protein
View	[ ]	chr1:1,000,000-1,000,000	LOC101927295	CDK2	CDK2 (transcript)	imprinted protein

Showing 1 to 5 of 294 entries

Select one row in the table above to see associated experiments for the selected gene

Targets based on the peak threshold of:

 All

 None

 Significant

 Labeled

 Network

All gene-related signals can be selected

Experiment table

Factor	Category	MeanScore	Association supported by	Classification	SEA Project	Study Title	Reference
ChIP-seq	ChIP-seq	0.177	ChIP-seq	ChIP-seq	MIP13349	Methicillin resistance system promoting A17 Genome sequencing and assembly	PubMed
SPOT	SPOT	0.163	None	None	MIP13349	Methicillin resistance system promoting A17 Genome sequencing and assembly	PubMed
Quantiflag	Quantiflag	-1	Significant	ChIP-seq	MIP13349	Methicillin resistance system promoting A17 Genome sequencing and assembly	PubMed
Hi-C	Hi-C	1.044	Labeled	Hi-C	MIP13349	Methicillin resistance system promoting A17 Genome sequencing and assembly	PubMed
RFC	RFC	0.172	Relevant	Hi-C	MIP13349	Methicillin resistance system promoting A17 Genome sequencing and assembly	PubMed

Showing 1 to 5 of 294 entries

Visualization in JBrowse

Signal tracks:

 BAM

 BAM

 BAM

 BAM

 BAM

 BAM

 Visualization in JBrowse:

Track color by: Default Factor Category

Track color style: Uniform Random ColorMap

Track height: 10

Additional tracks: GC percent, Mask Repeat

© 2019-2022 The ChIP-Atlas team. All rights reserved. This web application is maintained by the JGI Computational Biology Group.

5. For visitors who study specific molecular mechanisms, their purpose is to quickly obtain information such as the regulatory relationship of the gene(s) of interest. This platform didn't provide such services. The starting point of the analysis of this platform is based on the collected public data sets. There is no unified classification and integration of these data sets in advance, resulting in the lack of prominent functions and not attractive to users interested in biological mechanism.

> **Response:** As we replied the reviewer in our early response, we do provide such services for users to find regulatory relationship of the gene(s) of interest. The function is under the page of 'Browser' at the panel of 'Networks' (see a screenshot below).

6. For users who do big data analysis, the analysis often involves multiple data sets. The upper limit of a single analysis of this platform is only 10 data sets, which is inefficient and cannot meet the purpose of such users.

> **Response:** We thank the reviewer for the comment. It's indeed not feasible to perform big data analysis via the ChIP-Hub website since big data analysis always requires huge computing resources and more specific functions. However, for users who are interested in integration analysis of big data, ChIP-Hub provides rich resources by batch downloading for their downstream analysis. In order to help users to follow up the similar analysis as we did in the manuscript, we now provided the corresponding source code (it is under the link https://biobigdata.nju.edu.cn/ChIPHub_download/).

8. Instability. I have tried several analyses on this platform, but most of the time, I failed, which directly leads to disconnection from the server, making the user experience extremely poor.

> **Response:** We thank the reviewer to point this out. The current version of ChIP-Hub was developed based on the Shiny framework (<http://shiny.rstudio.com/>), which combines the computational power of R with friendly and interactive web interfaces. However, we used a FREE version of Shiny Server for our ChIP-Hub deployment. The website operation may not be friendly (e.g., dis-connection problem) and smooth enough (i.e., slow loading at the first time) in some way. This point has also been acknowledged by the reviewer in his/her early comments “..., *interactive functions such as page loading are not smooth enough. But I have to admit that the website built by R Shiny is indeed very beautiful, giving people a refreshing feeling ...*” The disconnection issue would be solved in a new version of ChIP-Hub interface based on a Python framework (under development).

Reviewer #2 (Remarks to the Author):

Thank Fu and colleagues for the response. It is very important in this era of omics to collect and re-analyze the omics data. In this manuscripts, Fu, et al collected > 10,000 datasets from > 40 plant species building a website that provides huge resource for future research. In this manuscript, some insight analysis is very useful for studying the relationship of TF and histone modifications like TF interaction regulatory network and tissue-specific regulatory elements etc. One of the most important research points for plants is how plants adapt to changing environments. If this part of data can be collected and analyzed in this research, it will make the paper more outstanding.

> **Response:** We would like to thank the reviewer for this very interesting point. Indeed, there are some open chromatin datasets treated by two different environmental stresses (including heat and dark). Using a similar analysis as we did for the tissue-specific regulatory elements, we identified a list of regulatory elements whose target genes are enriched for GO terms such as “response to heat”, “photosynthesis, light harvesting” and “cellular response to decreased oxygen levels” (see the Figure below; see also the **Supplementary Fig. 20** in the revision). We have added this part of analysis in the revised manuscript: ‘... *As an example, we analyzed public ATAC-seq data treated by*

several environmental stresses and identified a set of regulatory elements whose target genes are response to heat and dark (**Supplementary Fig. 20**). The results may provide novel insights into how plants adapt to changing environments.’

Supplementary Fig. 20. Dynamic activity of regulatory elements upon different stress treatments. (a) The boxplot showing the specificity score of promoter and enhancer

accessibility. All peaks (n=25,828) were included. **(b-c)** Analysis of dynamically accessible peaks (n=5,469) upon stress treatments. The cutoff of peak treatment specificity was set as Tau index < 0.05 or > 0.2 . **(b)** The optimal number (n=8) of clusters determined by an elbow method. **(c)** Heatmap showing the chromatin accessibility of 5469 highly specific regulatory elements (including promoters and enhancers). Regulatory elements are grouped into eight clusters based on their activity. Selected target genes are labeled on the right. **(d)** Boxplot showing the expression pattern of target genes in different clusters as in **(b)**. The gene expression data was downloaded from the Arabidopsis RNA-seq database (<http://ipf.sustech.edu.cn/pub/athrdb/>). **(e)** Genome browser views of treatment-specific chromatin accessibility and RNA-seq read intensity at the two chosen gene loci: *HSP70T-2* and *SOM*. **(f)** Network plot showing representative of enriched GO terms for target genes in different clusters. Only significantly enriched (adjusted p-value < 0.01) GO terms were shown. The circle size represents the number of genes belonged to in a specific term.

Reviewers' Comments:

Reviewer #1:

Remarks to the Author:

1. There are already a bunch of databases integrated similar tools and analyses, providing more in-depth analysis covering most of plant species. I can't find how much this work contributes to the community.

1) Similar datasets and analyses have been reported in PlantPAN3.0: a new and updated resource for reconstructing transcriptional regulatory networks from ChIP-seq experiments in plants. NAR 2019 (<http://PlantPAN.itps.ncku.edu.tw/>) . PlantPAN integrated TF ChIP-seq data from seven major model plants. ChIP-seq data from the seven species accounted for more than 90% of the datasets collected by ChIP-Hub.

2) The plant epigenomic data and cis-regulatory elements have also been processed and provided by multiple databases, and the chromatin state analysis has already been reported in PCSD: a plant chromatin state database. NAR 2018 (<http://systemsbiology.cpolar.cn/chromstates>), PlantGSAD: a comprehensive gene set annotation database for plant species (<http://systemsbiology.cpolar.cn/PlantGSEAv2/>).

3) In terms of the tools and integration of multi-omics data, there are already a large number of databases that provide more in-depth analysis and regulatory information. PlantRegMap (<http://plantregmap.gao-lab.org/>), which integrates transcriptional regulation information from 63 plant species and is regularly updated. A set of tools and packages are provided. Although the database does not provide online genome browsing, regulatory information is derived based on multiple genetic evidence including ChIP-seq profiles, conservation score and machine learning approaches, and the results are expected to be more robust.

2. In terms of data visualization, most ChIP-seq data in public databases are processed with peak file and bw files available, which can readily be visualized via uploading to genome browser, including online browser (<https://epigenomegateway.wustl.edu/browser/>) and integrated genome browser application (IGV).

3. The authors' response to the issue of data update and maintenance is too vague, 18 out of the 42 genome assemblies are already not up to date in ChIP-hub.

1) The authors mentioned "data update for a new gene model version is easy to work from peak files." However, when the genome is updated, read mapping and peak calling for all the datasets collected need to be redone.

2) With the rapid development of sequencing technologies, especially HIFI technology, genome updating has become more frequent, and the database updating and maintenance is a big problem.

3) It is true that genome coordinate could be transferred by 'liftOver', but the preparation of chain files for 'liftOver' is nontrivial. There is currently no liftOver chain files for plant species except arabidopsis. The authors need to demonstrate that they can provide online liftOver analysis for species between different versions.

4. There are already many online tools and platforms to quickly analyze public data, such as GALAXY, TBtools and "The ChIP-Seq tools and web server: a resource for analyzing ChIP-seq and other types of genomic data" (DOI:10.1186/s12864-016-3288-8), which provided more systematic analysis than those provided in ChIP-hub.

5. The title of 'regulome' is inappropriate. It is a 'plant ChIP-seq data browser' without systematic regulome for majority of the plant species. The regulatory network is only provided for about five species, and more than half of the species have no CRE annotation. While these information have already been well-organized and presented in many other databases including PlantGSAD, PCSD and PlantRegMap.

Reviewer #2:

Remarks to the Author:

The authors have added the comparison analysis of public ChIP-seq dataset about how plants adapt to the changing environments, which make the paper more significant. These additions in the revised manuscript have satisfied my previous suggests.

REVIEWER COMMENTS

Reviewer #1 (Remarks to the Author):

1. There are already a bunch of databases integrated similar tools and analyses, providing more in-depth analysis covering most of plant species. I can't find how much this work contributes to the community.

Response: We thank the reviewer again for his/her new but trivial comments. However, most of the comments reflect some misunderstandings / lack of insights or knowledge. Please find our detailed responses to every comment below.

1) Similar datasets and analyses have been reported in PlantPAN3.0: a new and updated resource for reconstructing transcriptional regulatory networks from ChIP-seq experiments in plants. NAR 2019 (<http://PlantPAN.itps.ncku.edu.tw/>). PlantPAN integrated TF ChIP-seq data from seven major model plants. ChIP-seq data from the seven species accounted for more than 90% of the datasets collected by ChIP-Hub.

Response: We do not agree with the reviewer that PlantPAN3.0 provides similar datasets and analyses with ChIP-Hub. Firstly, PlantPAN only includes fewer TF ChIP-seq datasets to generate matrices of TF binding sites at the sequence level but it does not provide peak, signal bw files and any other quality metrics, which are important to evaluate the binding intensity, tissue specificity and data quality. Secondly, ChIP-Hub provides other types of regulome data such as HM ChIP-seq and open chromatin data, which are useful to study the dynamics of chromatin states and enhancers / promoters. However, these datasets are not included in PlantPAN. Thirdly, the analyses between ChIP-Hub and PlantPAN are totally different. Our analyses are more focused on tissue-specific regulatory landscapes and conservation of regulatory elements (promoters and enhancers) as well as chromatin states among different plant species. Such analyses cannot be produced by PlantPAN.

2) The plant epigenomic data and cis-regulatory elements have also been processed and provided by multiple databases, and the chromatin state analysis has already been reported in PCSD: a plant chromatin state database. NAR 2018 (<http://systemsbiology.cpolar.cn/chromstates>), PlantGSAD: a comprehensive gene set annotation database for plant species (<http://systemsbiology.cpolar.cn/PlantGSEAv2/>).

Response: There are at least three aspects that our chromatin state analysis is different from PCSD. Firstly, chromatin states are known cell- or tissue-specific (doi: <https://doi.org/10.1038/nature09906>), we have put more effort into the investigation of chromatin states in a tissue-specific manner. However, the chromatin state in PCSD was defined in a pooled manner without tissue specificity. Secondly, we have analyzed the enrichment of different TF binding sites in different chromatin states to investigate the function of

chromatin states. This kind of analysis has not yet been performed in studies mentioned by reviewer. Thirdly, we have also investigated the conservation and divergence of chromatin in different plant species.

The PlantGSAD database is somehow not relevant to plant regulome in terms of content and function. We're not sure why did the reviewer mention this.

3) In terms of the tools and integration of multi-omics data, there are already a large number of databases that provide more in-depth analysis and regulatory information. PlantRegMap (<http://plantregmap.gao-lab.org/>), which integrates transcriptional regulation information from 63 plant species and is regularly updated. A set of tools and packages are provided. Although the database does not provide online genome browsing, regulatory information is derived based on multiple genetic evidence including ChIP-seq profiles, conservation score and machine learning approaches, and the results are expected to be more robust.

Response: We do not agree with the reviewer regarding to the comments. PlantRegMap only contains 280 datasets (see the Data Source section at <http://plantregmap.gao-lab.org/help.php>) in the analysis. Most regulation information stored in PlantRegMap is sequence-based TF binding motifs, which were generated predicted information for non-model species. We have collected more than 10000 datasets in ChIP-Hub (35 times more than PlantRegMap) which are experimental-based data with tissue- and condition-specific information.

2. In terms of data visualization, most ChIP-seq data in public databases are processed with peak file and bw files available, which can readily be visualized via uploading to genome browser, including online browser (<https://epigenomegateway.wustl.edu/browser/>) and integrated genome browser application (IGV).

Response: We integrated the WashU Epigenome Browser into ChIP-Hub not only because the official online browser (<https://epigenomegateway.wustl.edu/browser/>) does not include the used genome assemblies but also it provides an easy way for users to visualize interested datasets (users do not need to download and upload data files for visualization purposes).

3. The authors' response to the issue of data update and maintenance is too vague, 18 out of the of the 42 genome assemblies are already not up to date in ChIP-hub.

Response: We appreciate the reviewer letting us know that 18 out of 42 genome assemblies are not up to date. However, we have to mention that big data collection, uniform processing and reuse remain a big issue in plant research but such attempt is still limited so far. Such kind of work is very labor-expensive and time-consuming and generally requires consortium effort. Our work offers a starting point to fill this

gap. The reviewer has to admit that, even for the consortium-based projects such as ENCODE and TCGA, data update does not keep pace with genome update. We also call for consortium efforts to take over ChIP-Hub-like work in the future so that more and more plant scientists can benefit from it.

1) The authors mentioned “data update for a new gene model version is easy to work from peak files.” However, when the genome is updated, read mapping and peak calling for all the datasets collected need to be redone.

Response: We thank the reviewer to point out the same issue again. There are several different solutions to solve the issue. Firstly, if there is a new major version of genome assembly for a specific plant species by the plant community, we may consider providing another release by reanalysis of all the datasets in the plant species. Secondly, since most genes are stable in different assemblies, users can still use the ChIP-Hub data when their analyses are gene-based. We would provide a gene ID convertor for this in our new version of ChIP-Hub. Last but not least, genome coordinates can be transferred by “liftOver” for new genome versions (see our Response 3.3 below).

2) With the rapid development of sequencing technologies, especially HIFI technology, genome updating has become more frequent, and the database updating and maintenance is a big problem.

Response: We appreciate the reviewer recognizing the potential challenges for data update. However, this does not necessarily mean that it’s not doable. As we mentioned above, there are already known solutions for this.

3) It is true that genome coordinate could be transferred by ‘liftOver’, but the preparation of chain files for ‘liftOver’ is nontrivial. There is currently no liftOver chain files for plant species except arabidopsis. The authors need to demonstrate that they can provide online liftOver analysis for species between different versions.

Response: We also recognized that there were no comprehensive liftOver chain files in plants. However, these are some kinds of work that we are doing and are going to do. For example, we have shown that it’s possible to implement such functions in the model species of rice (doi: <https://doi.org/10.1016/j.jgg.2022.04.003>). In the near future, we would apply this to other plant species in our ChIP-Hub platform.

4. There are already many online tools and platforms to quickly analyze public data, such as GALAXY, TBtools and “The ChIP-Seq tools and web server: a resource for analyzing ChIP-seq and other types of

genomic data” (DOI:10.1186/s12864-016-3288-8), which provided more systematic analysis than those provided in ChIP-hub.

Response: It’s true that these online tools can be used for ChIP-seq data analysis. However, these platforms do not provide ready-to-use data sources. ChIP-Hub is especially useful when scientists would like to promptly check the potential target sites of a specific regulator without troubling for reanalyzing the data.

5. The title of ‘regulome’ is inappropriate. It is a ‘plant ChIP-seq data browser’ without systematic regulome for majority of the plant species. The regulatory network is only provided for about five species , and more than half of the species have no CRE annotation. While these information have already been well-organized and presented in many other databases including PlantGSAD, PCSD and PlantRegMap.

Response: ChIP-Hub offers a reference database to provide comprehensively reanalyzed regulatory genomic data (regulome) that have been produced by the whole plant community so far. The regulatory network and CRE information would be available for more plant species as more and more regulome data are generated by the plant community. We have changed our manuscript title to “ChIP-Hub Provides an Integrative Platform for Exploring Plant Regulome”.